



# dh2loop 1.0: an open-source python library for automated processing and classification of geological logs

Ranee Joshi[1,2], Kavitha Madaiah[1,2], Mark Jessell[1,2], Mark Lindsay[1,2] and Guillaume Pirot[1,2]

[1]Centre of Exploration Targeting, School of Earth Sciences, University of Western Australia, 35 Stirling Highway, Crawley 6009 Western Australia
[2]Mineral Exploration Cooperative Research Centre (MinEx CRC), School of Earth Sciences, University of Western Australia, 35 Stirling Highway, Crawley 6009 Western Australia

*Correspondence to*: Ranee Joshi (ranee.joshi@research.uwa.edu.au)

**Abstract.** Exploration and mining companies rely on geological drill core logs to target and obtain initial information on geology of the area to build models for prospectivity mapping or mine planning. A huge amount of legacy drilling data is available in geological survey but cannot be used directly as it is compiled and recorded in an unstructured textural form and using different formats depending on the database structure, company, logging geologist, investigation method, investigated materials and/or drilling campaign. It is subjective and plagued with uncertainty as it is likely to have been conducted by tens to hundreds geologists, all of whom would have their own personal biases. However, this is valuable information that adds value to geoscientific data for research and exploration, specifically in efficiently targeting sustainable new discoveries and providing better shallow subsurface constraints for 3D geological models.

*dh2loop* (https://github.com/Loop3D/dh2loop) is an open-source python library that provides the functionality to extract and standardize geologic drill hole data and export it into readily importable interval tables (collar, survey, lithology). In this contribution, we extract, process and classify lithological logs from the Geological Survey of Western Australia Mineral Exploration Reports Database in the Yalgoo-Singleton Greenstone Belt (YSGB) region. For this study case, the extraction rate for collar, survey and lithology data is respectively 93%, 865 and 34%. It also addresses the subjective nature and variability of nomenclature of lithological descriptions within and across different drilling campaigns by using thesauri and fuzzy string matching. 86% of the extracted lithology data is successfully matched to lithologies in the thesauri. Since this process can be tedious, we attempted to test the string matching with the comments, which resulted to a matching rate of 16% (7,870 successfully matched records out of 47,823 records). The standardized lithological data is then classified into multi-level groupings that can be used to systematically upscale and downscale drill hole data inputs for multiscale 3D geological modelling. *dh2loop* formats legacy data bridging the gap between utilization and maximization of legacy drill hole data and drill hole analysis functionalities available in existing python libraries (*lasio*, *welly*, *striplog*).





## 1 Introduction

Drilling is the process of penetrating through the ground and extracting rocks from various depths beneath the surface for confirming the geology beneath and/or providing samples for chemical analysis. As it penetrates the ground it forms drill holes

from which drill core is collected. The location of where drilling starts is referred to as the collar. As the drilling progresses, survey orientation measurements are taken to be able to convert the specific depths to exact coordinate locations of the drill core being retrieved. In a hard rock setting, geological drill core logging is the process whereby the recovered drill core sample is systematically studied to determine the lithology, mineralisation, structures, and alteration zones of a potential mineral deposit. It is usually performed by geologists who classify a rock unit into a code, based on one or multiple properties such as

rock type, alteration intensity and mineralisation content. Exploration and mining companies rely on the diverse geoscientific information obtained by drill core logging techniques to target and to build models for prospectivity mapping or mine planning. This work focuses on lithological logs which is the component of a geological log that refers to the geological information on the dominant rock type in a specific downhole interval. Inevitably, lithological drill core logging is subjective and plagued with uncertainty, particularly as it is likely to have been conducted by ten to hundred geologists, all of whom would have their

own personal biases (Lark et al., 2014). Furthermore, it can be difficult to recognize lithology with confidence and to establish subtle variations or boundaries in apparently homogeneous sequences.

With the advent of the digital age, semi-automated drill core logging techniques such as X-Ray Diffraction (XRD), X-Ray Fluorescence spectrometry (XRF) and Hyperspectral (HS) imaging have provided higher detail of data collection and even

detection of other properties such as conductivity, volumetric magnetic susceptibility, density using gamma-ray attenuation, and chemical elements during logging (Zhou et al., 2003;Rothwell and Rack, 2006;Ross et al., 2013).  This has prompted a shift towards using numerical data rather than depending on traditional geological drill core logging procedures (Culshaw, 2005). Multiple methods have been recently applied to geological drill core logging such as wavelet transform analysis or data mosaic (Arabjamaloei et al., 2011;Hill et al., 2020;Le Vaillant et al., 2017;Hill et al., 2015),  artificial neural network model

(Lindsay, 2019;Zhou et al., 2019;Emelyanova et al., 2017) and inversion (Zhu et al., 2019). Relying solely on these semi-automatic methods comes with drawbacks as it excludes some of the subjective interpretations that cannot be replaced. Furthermore, a rich amount of legacy data was collected in the traditional drill core logging method and disregarding this information limits the dataset.

Legacy data are information collected, compiled and/or stored in the past into many different old or obsolete formats or systems, such as handwritten records, aperture cards, floppy disks, microfiche, transparencies, magnetic tapes and/or newspaper clippings making it difficult to access and/or process (Smith et al., 2015). In geoscience, these are currently scattered amongst unpublished company reports, departmental reports, publications, petrographic reports, printed plans and maps, aerial photographs, field notebooks, sample ticket books, drill core samples, tenement information and geospatial data providing a

major impediment to their efficient use. This includes geological drill core logs that are the outcome of most expensive part of most mineral exploration campaigns: drilling. This is valuable information source and key assets that can be used to add value to geoscientific data for research and exploration; design mapping programs and research questions of interest; more efficiently target remapping and sustainable new discoveries; and provide customers with all existing information at the start of the remapping program. It should not be abandoned for it may have lower intrinsic quality than observations made with more

modern equipment, its recovery and translation to a digital format is too tedious. Elizabeth Griffin (2015) argues that there is no distinction in principle between legacy data and 'new' data, as all of it is data.  The intention of recovering legacy data is to a) upcycle information with integration into modern datasets, b) use salvaged data for new scientific applications and c) allow reuse of that information into utility downstream applications (Vearncombe et al., 2017). Furthermore, extracting





information from legacy datasets is high and relatively low-risk as geoscientific insight is added to a project for little or no
cost compared to those of drilling (Vearncombe et al., 2016).

The primary challenge in dealing with geological legacy datasets is a large amount of important data, information and
knowledge are recorded in an unstructured textural form, such as host rock, alteration types, geological setting, ore-controlled
factors, geochemical and geophysical anomaly patterns, and location (Wang and Ma, 2019). Moreover, the geological drill
core logging forms and formats vary depending on the company, logging geologist, investigation method, investigated
materials and/or drilling campaign. Natural language processing (NLP) also known as computational linguistics has been used
for information extraction, text classification and automatic text summarization. NLP relies on data-driven computation
involving statistics, probability, machine learning and "deep learning" (Otter et al., 2020). NLP applications on legacy data
have been demonstrated in the fields of  taxonomy (Rivera-Quiroz and Miller, 2019), biomedicine (Liu et al., 2011) and legal
services (Jallan et al., 2019). Qiu et al. (2020) proposed an ontology-based methodology to support automated classification
of geological reports using word embeddings, geoscience dictionary matching and bidirectional long short-term memory model
(Dic-Att-BiLSTM) that assists in identifying the difference in relevance from a report.  Padarian and Fuentes (2019) also
introduced the use of domain-specific word embeddings (GeoVec) which was used to automate and reduce subjectivity of
geological mapping of drill hole descriptions (Fuentes et al., 2020).


Similarity matching has many applications in natural language processing as it is one of the best techniques for improving
retrieval effectiveness (Park et al., 2005). The use of text similarity is beneficial for text categorization (Liu and Guo, 2005)
and text summarization (Erkan and Radev, 2004;Lin and Hovy, 2003). Fuzzy string matching, also known as approximate
string matching, is the process of finding strings that approximately match a given pattern (Cohen, 2011;Gonzalez et al., 2017).
It has been used in language syntax checker, spell-checking, DNA analysis and detection, spam detection, sport and concert
event ticket search (Higgins and Mehta, 2018), text re-use detection (Recasens et al., 2013) and clinical trials (Kumari et al.,
2020).

Most of the python libraries available have been built to process extracted and standardized drill hole data. The most common
of these are: *lasio* (https://lasio.readthedocs.io/en/latest/) which deals with reading and writing Log ASCII  Standard (LAS)
files, a drill hole format commonly used in the oil and gas industry, *welly* (https://github.com/agile-geoscience/welly) which
deals with loading, processing, and analysis of drill holes and *striplog* (https://github.com/agile-geoscience/striplog) which
digitizes, visualizes and archives stratigraphic and lithological data. *Striplog* (Hall and Keppie, 2016) also parses natural
language 'descriptions', converting them into structured data via an arbitrary lexicon which allows further querying and analysis
on drill hole data. The main limitations of these existing libraries, with respect to legacy data in the mining sector is that they
assume that the data is already standardized and pre-processed.

*dh2loop* provides the functionality to extract and standardize geologic drill hole data and export it into readily importable
interval tables (collar, survey, lithology). It addresses the subjective nature and variability of nomenclature of lithological
descriptions within and across different drilling campaigns by integrating published dictionaries, glossaries and/or thesauri
that were built to improve resolution of poorly defined or highly subjective use of terminology and idiosyncratic logging
methods. It is however important to highlight that verifying the accuracy and/or correctness of the geological logs being
standardized is outside the scope of this tool, thus we assume logging has been conducted to the best of the geologist's ability.

Furthermore, it classifies lithological data into multi-level groupings that can be used to systematically upscale and downscale
drill hole data inputs in multiscale 3D geological model. It also provides drill hole desurveying (computes the geometry of a



drillhole in three-dimensional space) and log correlation functions so that the results can be plotted in 3D and analysed against each other. It also links the gap between utilization and maximization of legacy drill hole data and the drill hole analysis functionalities available in existing python libraries.

**2 Materials and Methods**

**2.1 Conventions**

This paper involves multiple python libraries, database tables and fields. For clarity, the following conventions are used for this paper:

1. Python libraries are written in italics: *dh2loop*
2. Python functions are written in italics followed by an open and close parenthesis: *token_set_ratio()*
3. Database tables are written in Lucinda Console Italics: `dhgeology`
4. Database table fields are written in Lucinda Console: `CollarID`

**2.2 Dependencies**

*dh2loop* stands for drill hole data extracted into a 3D modelling input format, compatible with/for the Loop platform. It is a
drill hole processing tool that integrates published dictionaries, glossaries and/or thesauri to and improve standardize highly subjective use of terminology and idiosyncratic logging methods and classify lithological logs. It primarily depends on a number of external open-source libraries:

1. *fuzzywuzzy* (https://github.com/seatgeek/fuzzywuzzy) which uses fuzzy logic for string matching (Cohen, 2011)
2. *pandas* (https://pandas.pydata.org/) for data analysis and manipulation (McKinney, 2011)
3. *psycopg2* (https://pypi.org/project/psycopg2/), a PostgreSQL database adapter for python (Gregorio and Varrazzo, 2018)
4. *numpy* (https://github.com/numpy/numpy)
5. *nltk* (https://github.com/nltk/nltk ), the Natural Language Toolkit is a suite of open source Python modules, data sets, and tutorials supporting research and development in Natural Language Processing (Loper and Bird, 2002).
6. *pyproj* (https://github.com/pyproj4/pyproj), python interface to PROJ (cartographic projections and coordinate transformations library)

Code describing basic drill hole operations, such as desurveying (process of translating collar (location) and survey data (azimuth, dip, length) of drill holes into XYZ coordinates in order to define its 3D geometry of the non-vertical borehole), was heavily inspired from *pyGSLIB* drill hole module (Martínez-Vargas, 2016). *pyGSLIB* (https://github.com/opengeostat/pygslib)
is an open-source python package to perform mineral resource estimations. The *pyGSLIB* drillhole module handles drill hole data, desurveying interval tables and other drill hole related processes. The module was re-written into python to be make it more compact with less dependencies and tailor it to the data extraction output.

**2.3 Data Source**

The Geological Survey of Western Australia Mineral Exploration Reports Database contains open-file reports submitted as a
compliance to the Sunset Clause, Regulation 96(4) of the Western Australia legislation Mining Regulations 1981. These reports contain valuable exploration information in hardcopy (1957-2000), hardcopy and digital format (2000-2007) and digital format (2000-present) (Riganti et al., 2015). The minimum contents of a drilling report comprise a collar file which describe the geographic coordinates of the collar location (Fig. 1). Additional files may be included, such as a survey file describing the depth, azimuth and dip measurements for the drilling path; assays; downhole geology and property surveys (e.g. downhole
geochemistry, petrophysics) may also be available depending on the company's submission (Riganti et al., 2015). The data in





the drilling reports were extracted with spatial attribution and imported to a custom-designed relational database (also called the Mineral Drillhole Database) curated by the GSWA that allows easy retrieval and spatial querying. For simplicity, we will refer to this database as the WAMEX database in this text.

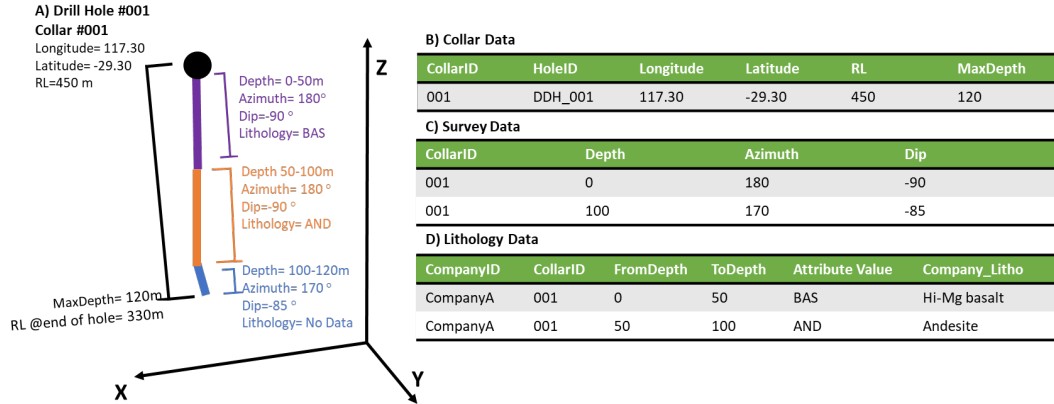

**Figure 1.** Simplified example of a drill hole (1.A) and its corresponding interval tables collar (1.B), survey (1.C) and lithology (1.D). The black circle denotes the collar location of the drill hole which is obtained from a collar table (1.B). The purple line represents the first downhole interval taking its deviation data from the survey table (1.C) and the lithology information from the lithology table (1.D). The same applies for the second interval (orange line) and third interval (blue line). The orange line follows the same trajectory as the first interval as it uses the same entry in the survey table (1.C). The blue line has no lithology data as this information is not present in the lithology table (1.D). The `MaxDepth` denotes the total drill length (1.B).

Each drill hole in the WAMEX database is identified by its surface coordinates and a unique ID (`CollarID`) in the *collar* table. The drill hole 3D geometry is described in the survey tables (*dhsurvey, dhsurveyattr*). The properties logged are described in pair of tables such as *dhgeology, dhgeologyattr*. The WAMEX schema is linked by a main table, *collar*, which assigns a primary key (`CollarID`) for each drill hole in Western Australia and corresponding geographic location (Fig. 2). *collar* has a one-to-many relationship with another table, *collarattr*, to store other attributes that describe each unique drill hole, such as maximum depth and elevation. The deviation of the drill hole is stored in a table, *dhsurvey*, with a primary key (`DHSurveyID`) that refers to each unique depth of a drill hole. This primary key has a many-to-one relationship with *collar*, as there are multiple depth measurements for each drill hole. Furthermore, *dhsurvey* also has a one-to-many relationship with table *dhsurveyattr*, which stores additional attribute information regarding survey, such as azimuth and dip readings. This similar relationship is maintained with interval tables, except that the primary key (e.g.`DHGeologyID`) is used to refer a unique downhole interval rather than a depth measurement. For lithological information, we refer to tables: *dhgeology* and *dhgeologyattr*. *dhgeologyattr* which contain information such as rock names and free text comments while *dhgeology* provides information to which hole and interval depth that data refers to. This information can be joined and extracted through SQL (Structured Query Language) queries. Other interval tables available in the database refer to typical logs and downhole surveys: alteration, events, geochemistry geophysics, geotech, hyperspectral, magnetic susceptibility, mineralogy, recovery, regolith, specific gravity, structure, veining, water level and weathering information.



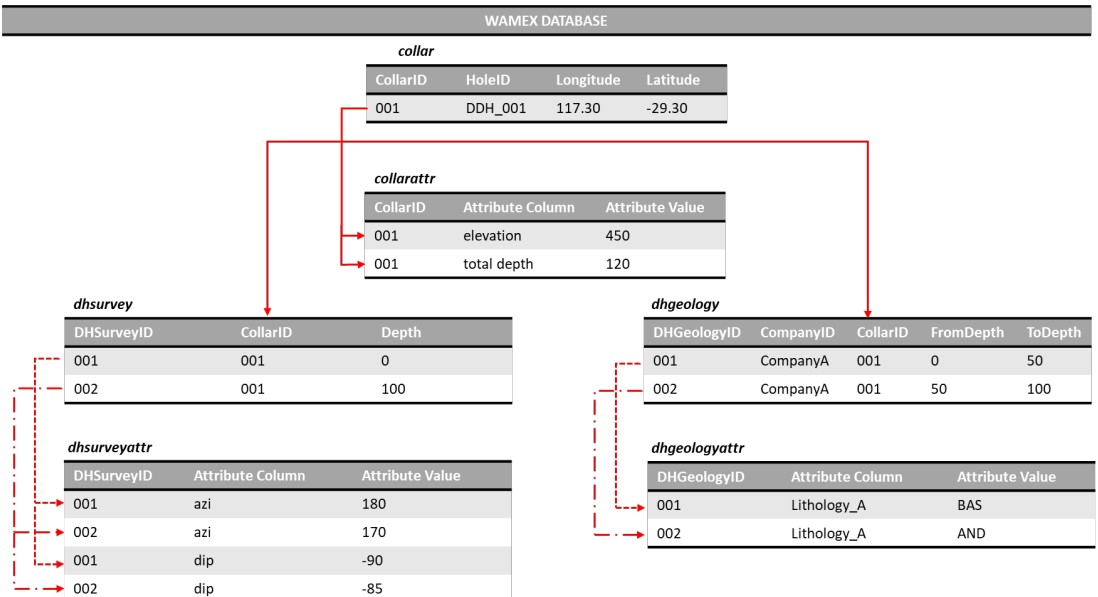

**Figure 2. Simplified WAMEX database schema showing the one-to-many relationship between the _collar_ table and the _collarattr_ table (red solid line). It also shows the relationship between the collar table and the other interval tables such as _dhsurvey, dhsurveyattr, dhgeology, dhgeologyattr_. The example show the relationship between tables for the first (red dashed line) and second interval (red dashed-dot line).**

It contains more than 50 years worth of mineral exploration drill hole data with more than 2.05 million drill holes, imported from over 1,514 companies. Each drill hole is identified by its surface coordinates and its unique ID (CollarID). The drill hole 3D geometry is described in the survey tables (_dhsurvey, dhsurveyattr_). However, it is important to emphasize that the drill hole data is of variable quality and reliability and no validation has been done. The amendments and reformatting necessary to be able to extract and utilize data from the WAMEX database is part of the functionalities provided by _dh2loop_.

## 2.4 Study Area

In this paper, we demonstrate the application of _dh2loop_ to data from the Yalgoo-Singleton greenstone belt (YSGB) (Fig. 2), a geologically complex, largely heterogeneous and highly mineralized arcuate granite-greenstone terrane, in the western Youanmi Terrane, Yilgarn Craton in Western Australia (Anand and Butt, 2010). The YSGB has good range of different lithologies in the area. Igneous rocks occur as extensive granitoid intrusions that occurred between 2700 and 2630 Ma (Myers, 1993), as well as ultramafic mafic volcanic rocks formed as extensive submarine lavas and local centres of felsic and mafic volcanic rocks. Some layered gabbroic sills intruding the greenstone are also observed. Sedimentary rocks formed in broad basins during tectonic and volcanic quiescence consist of mostly banded iron formation (BIF) and felsic volcanoclastic rocks. The greenstone belt is metamorphosed to greenschist facies (Barley et al., 2008). The area is also covered by deeply weathered regolith profiles which both conceals mineral deposits hosted by the underlying bedrock and contains signatures of mineralisation that are distal signatures of possible economically significant deposits (Cockbain, 2002). Furthermore, the YSGB is a major target for exploration as it has considerable resources of gold, nickel, bauxite, as well as lesser amounts of a wide range of other commodities (Cockbain, 2002). It hosts multiple mineral deposits ranging from volcanogenic massive sulphide (Golden Grove, Gossan Hill), orogenic gold (Mt. Magnet), banded iron formations (Mount Gibson, Karara, Extension Hill). The geological complexity and relevance to mineral exploration makes the YSGB a reasonable and sensible area to test the dh2loop thesauri, matching and upscaling.




**Figure 3.** The map shows the Yalgoo-Singleton greenstone belt highlighting the different mines and prospects in the area. The inset map shows the heterogeneous distribution and drill hole density from the legacy data available from the WAMEX database.





## 2.5 Thesauri

Since most exploration companies have their own nomenclature and systems, which could also change between drilling
campaigns, it is necessary to build thesauri: dictionaries that list equivalent and related nomenclature (or synonyms) for different attribute names and values. These thesauri are stored as additional tables in the database. For example, if we are interested in the major lithology in a specific interval, this information can be tabulated as "Major Rock Type", "Lithology_A" or "Main_Geology_Unit" depending on the drill core logging system used. The resulting thesauri considers change in cases, abbreviations, addition of characters, typographical errors and a combination of these. Although listing out these terms is
manual and tedious, it only needs to be done once and can be re-used and forms the basis for future text matching and as a training set to automate finding similar terms. This was preferred over selection based on regular expressions as when parsing these terms, there are complex patterns in the terms used and the inconsistencies in the way they are written that can be understood by a person with a geological background but not by a simple regular expression. The complexity of the regular expression required to catch all the terms of interest means an optimal expression is difficult, if not impossible, to define, and
also tends to be computationally burdensome. dh2loop 1.0 provides several thesauri that can easily be updated (if needed) for the following attributes (Appendix B: Thesauri):

1. Drill hole collar elevation (Appendix B1): 360 synonyms such as "elevation" and "relative level"
2. Drill hole maximum depth (Appendix B2):160 synonyms such as "end of hole", "final depth" and "total depth"
3. Drill hole survey azimuth (Appendix B3): 142 synonyms
4. Drill hole survey dip (Appendix B4): 8 synonyms such as "inclination"
5. Drill hole lithology (Appendix B5): 688 synonyms such as "geology", "Lithology_A", "Major_Geology_Unit" and "Major_Rock_Type"
6. Drill hole comments (Appendix B6): 434 synonyms such as "description"

The thesauri created specifically for further processing lithology and comments information are:
7. Drill hole lithology codes thesaurus (Appendix B7, discussed further in Sect. 2.5.*1*)
8. Clean up dictionary (Appendix B8, discussed further in Sect. 2.5.2)
9. Lithology hierarchical thesaurus (Appendix B9, discussed further in Sect. 2.5.*3*)

In order to extract the other attributes we envisage developing other thesauri, following the same workflow.

### 240   2.5.1   Drill hole lithology codes

This is a thesaurus compiling the equivalent lithology for a given lithological code based on the reports submitted to GSWA. This thesaurus is identified by a company id and report number. The current thesaurus covers 41 out of the 168 companies in the study area with a total of 352 entries (Appendix B7). It is important to note that the `Company_LithoCode` varies depending on the `CompanyID`. For example, "Company 551" refers to "Saprolite" as "CS" while Company "2551" uses CS
to refer to "Cambrian Sediment". The opposite is true as well, a company may use "AMPH" to refer to "Amphibolite" while another company may use "MAA".

### 2.5.2   Clean-up Dictionary

The clean-up dictionary is a list of words and non-alphabetic characters that are used as descriptions in the geological logging
syntax. This dictionary is used to remove these terms from the lithology and/or comment free text descriptions prior to the fuzzy string matching. The dictionary is composed of 1662 records, most of which were compiled from abbreviations in field and mine geological mapping (Chace, 1956) and the CGI vocabularies: GeoSciML and EarthResourcesML (Simons et al., 2006;Richard et al., 2007;Raymond et al., 2012). 353 of these records are original to *dh2loop* and were added to accommodate the geological logging syntax in Western Australia. Added records include the following:
1. Chronostratigraphic ages (Cambrian, Proterozoic)



2. Location descriptors (above, below, between), relative time (ancient, older, youngest)

3. Structural descriptors (anticlinal)

4. Textures (rounded, angular, block)

5. Mineralisation-related terminologies (absent, massive, disseminated)


6. Minerals (bornite, cassiterite),

7. Colors (brownish, cream)

8. Adjectives and their root form (good, better, best, extremely, extreme, fragmental, fragments, fragment)

9. Symbols (>, ?,\,@);  and

10. Common phrases (same as above, as per usual).


### 2.5.3    Lithology Hierarchical Thesaurus

The lithology hierarchical thesaurus is a list of 757 rock names (`Detailed_Lithology`), their synonyms and a two-level upscale grouping (`Lithology_Subgroup` and `Lithology_Group`). Each row in `Detailed_Lithology` refers to a rock name. Each rock name row lists the standardized terminology first, followed by its synonyms. The two corresponding

columns for this row indicated the two-level upscale grouping. 169 of these rock names were compiled from GeoSciML (Simons et al., 2006;Richard et al., 2007;Raymond et al., 2012). The synonyms were obtained from mindat.org (Ralph and Chau, 2014;Ralph, 2004).  The hierarchical classification was inherited from both mindat.org (Ralph and Chau, 2014;Ralph, 2004) and the British Geological Survey (BGS) Classification Scheme (Gillespie and Styles, 1999;Robertson, 1999;Hallsworth and Knox, 1999;McMillan and Powell, 1999;Rosenbaum et al., 2003). It is important to use multiple libraries to be able to

build an exhaustive thesauri as some libraries are limited by the nomenclature, level of interest and presence of the lithology or rock group in a geographic area. For example, the BGS classification did not have a comprehensive regolith dictionary, but one could argue that this is because the regolith is not extensive and not known to be a host for mineralisation in the United Kingdom. Thus, regolith has been classified using the regolith glossary (Eggleton, 2001).

The hierarchical classification consists of seven major lithology groups (`Lithology_Group`) (Fig. 4):

1) Igneous rocks. The igneous rocks are further classified to 12 lithology subgroups, considering grain size, composition and a combination of both.

2) Sedimentary rocks. Sedimentary rocks sub-classified to 16 lithology subgroups based on genetic source and composition (carbonate, clastic, evaporate, hybrid, hydroxide, ironstone, non-clastic siliceous, organic-rich,

phosphate, siliceous, siliciclastic, volcaniclastic, glacigenic).

3) Metamorphic rocks. Metamorphic rocks are subdivided into nine lithology subgroups based on the degree and type of metamorphism (metasomatic, contact, low-grade, schist, gneiss, high-grade, granofels, greenschist, metacarbonate).

4) Surficial rocks. Surficial rocks are subdivided into 13 lithology subgroups based on the depositional environment and

composition. The residual deposit `Lithology_Subgroup` includes the regolith detailed lithologies.

5) Mineralisation.  Mineralisation is considered as a separate classification to be able to classify ore zones.

6) Structure and texture. Since structures and textures can sometimes be logged as lithologies in geological logging, they are classified separately. Structure and Texture is divided into five lithology subgroups: fault rock, breaks, contact, fillings and sedimentary structures. And

7) Unclassified. The final classification is a catch–all for unclassified rocks.





**Figure 4. Lithology hierarchical thesauri showing the 7 major `Lithology_Groups` and their corresponding `Lithology_Subgroups`:** Igneous rocks (pink), Sedimentary rocks (light brown), Metamorphic rocks (green), Surficial Rocks (light yellow), Texture and Structure (blue), Mineralisation (purple) and Unclassified Rocks (dark yellow). Igneous rocks `Lithology_Subgroups` considers grain size, composition and a combination of both. Sedimentary rocks are sub-classified based on genetic source and composition. Metamorphic rocks are subdivided based on the degree and type of metamorphism. Surficial rocks are subdivided based on the depositional environment and composition. Mineralisation is considered as a separate



**classification to be able to classify ore zones. Structure and texture addresses situations that structures are logged as lithologies in geological logging. The final classification is a catch–all for unclassified rocks.**





### 2.4 Data Extraction

Currently, the *dh2loop* library extracts collar, survey and lithology information. It uses a configuration file (Appendix C1) that

allows the user to define the inputs, which are:

1. Region of interest (in WGS 1984 lat/long); and/or
2. List of drill hole ID codes codes, if known.
3. If reprojection is desired, the EPSG code of the projected coordinate system (e.g. EPSG:28350 for MGA Zone 50; http://epsg.io)

4. The connection credentials to the local copy of the WAMEX database
5. Input and output file directories/locations

### 2.4.1 Collar Extraction

With the minimum input of a region of interest, the *dh2loop* library exports a Comma-Separated Values file (CSV) listing the drill holes in the area with the following information (Fig. 5):

1. `CollarID`: This is the primary key from the *collar* table. It is used to associate data in different tables with a single drill hole. The `CollarID` for a drill hole is identical in all tables in order for data to be associated with that drill hole.

2. `HoleID`: This is the drill hole name, as the company would internally identify the drill hole.

3. `Longitude` and `Latitude`: The geographical coordinates locating the collar of the drill hole. Both values are

expressed in WGS 1984 lat/long (EPSG:2436).

4. Relative level (`RL`): This refers to the Z coordinate of the collar location. This value is extracted by using the drill hole collar elevation thesaurus to filter the values referring to relative level (Fig. 5b). More than one value can be fetched due to duplicate company submissions or multiple elevation measurements, in which case the code retains the value with most decimal places assuming higher precision corresponds to better accuracy. If no elevation values

are fetched from the database the entire record is skipped. Non-numeric values are also ignored.

5. Maximum depth (`MaxDepth`): This refers to the maximum downhole length drilled for a drill hole, commonly referred as the end-of-hole. This value is extracted by using the drill hole collar maximum depth thesaurus (Fig. 5c). Due to duplicate company submissions, there can be more than one value fetched. Since there is no submission date information, the code takes the value with largest value assuming it is the latest submission.

6. Calculated `X`, `Y` values of projected coordinates: These values are commonly calculated and used to be able to plot the drill hole in a metric system to be able to accurate display and measure distance within and between drill holes. The projection system used in the calculation is based on the input specified in the configuration file (Sect. 2.4, Appendix C1)



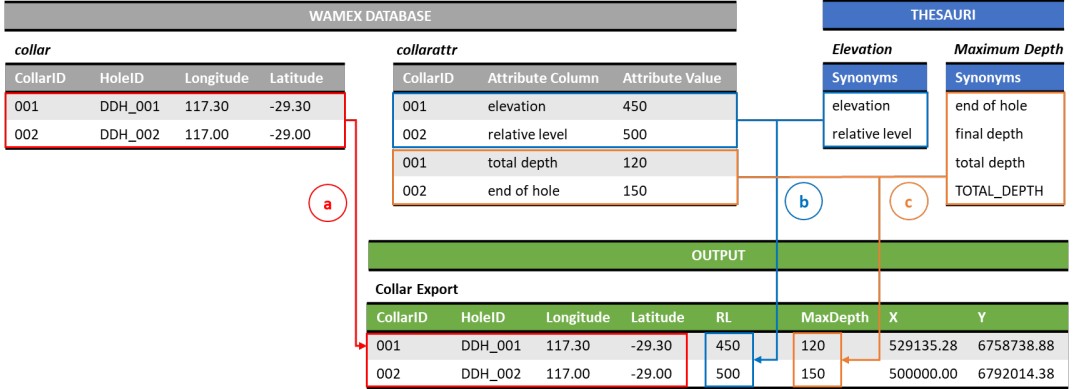

**Figure 5. Collar extraction workflow showing the `CollarID`, `HoleID`, `Longitude` and `Latitude` information is fetched from the `collar` table (a, red), the corresponding `RL` and `MaxDepth` values are fetched from the `collarattr` table using the Elevation (b, blue) and Maximum Depth thesauri (c, orange).**

**2.4.2 Survey Extraction**

With the same inputs defined in the configuration file, the *dh2loop* library outputs a survey CSV file containing the following

information (Fig. 6):

1.  `CollarID`: The primary key to link the survey information to the collar file.

2.  `Depth`: This is from the survey table and refers to the downhole depth where the survey measurement was taken. All non-numeric values were ignored. If the depth value fetched is negative, the absolute value is taken as it may have been used to denote direction of drilling. This assumption was made as some drill holes have negative depth

information that is technically not possible to have a negative length. This was done by some companies to denote that the depth measure was going upwards (usually for underground probing drill holes) rather than downhole. To put all the drill holes in the same cartesian plane, this correction was done.

3.  `Azimuth`: It is the trend direction indicated by an angle between 0-360 degrees from the north going clockwise. This is extracted using the drill hole azimuth thesaurus (Fig. 6b). The code fetches values between 0-360 degrees, thus

ignoring non-numeric value and values greater than 360. Values between -360 to 0 are assumed to be counter-clockwise from the north. If there is no survey information for a drill hole present in collar, the azimuth value is set to 0.

4.  `Dip`: It is the inclination angle perpendicular to the azimuth indicated by an angle between -90 to 90. It is measured from the horizontal plane, thus a positive value would describe a drill hole deviating to the surface or points upward

while a negative value would be a drill hole pointing downwards. This is extracted using the drill hole dip thesaurus (Fig. 6c).

5.  Calculated `X`, `Y`, `Z` values: These values are the project location of the survey measurement. This is calculated using the minimum curvature (also called spherical arc) algorithm. The minimum curvature algorithm (Amorin, 2009) desurveys downhole distances as distances along a circular arc. The algorithm matches the survey at two consecutive

measurements exactly and the curvature is constant between these two measurements. The direction remains continuous meaning there are no sharp changes in direction. The code for the minimum curvature was based on the *pyGSLIB* drill hole module.



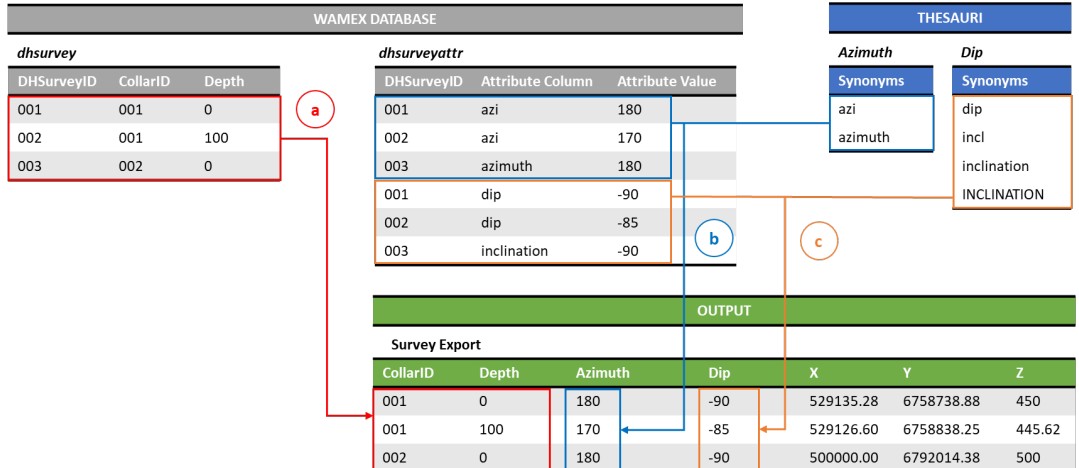

**Figure 6. Survey extraction workflow showing the `DHSurveyID`, `CollarID` and `Depth` information is fetched from the _dhsurvey_ table (a, red), the corresponding Azimuth and Dip values are fetched from the _dhsurveyattr_ table using the Azimuth (b, blue) and Dip thesauri (c, orange).**

### 2.4.3 Lithology Extraction

The lithology extraction outputs a lithology CSV file containing the following information (Fig. 7):

1. `CompanyID`: The primary key to link the lithology code to the drill hole lithology codes thesaurus and decode the lithologies.

2. `CollarID`: The primary key to link the lithology information to the collar file.

3. `FromDepth` and `ToDepth`: The start/from and end/to downhole depth values. If the `ToDepth` is null, we assume `ToDepth` to be equal to `FromDepth` + 0.01. If the `FromDepth` is larger than `ToDepth`, the `FromDepth` and `ToDepth` values are switched.

4. `Detailed_Lithology`: This value is the lithology matched through fuzzy string matching. The string that serves as input to the fuzzy string matching may either be the `Company_Litho` (decoded lithology from `Company_LithoCode`) or from the `Comments` (free text descriptions).

   4.1. Decoding Lithological Codes

      4.1.1. `Company_LithoCode`: This fetches the lithology codes that are typically three-letter codes using the drill hole lithology thesaurus.

      4.1.2. `Company_Litho`: The Company_Litho is fetched by matching the CompanyID and Company_LithoCode to the drill hole lithology codes thesaurus.

   4.2. `Comments`: This fetches the free text descriptions using the drill hole comments thesaurus.

5. `Lithology_Subgroup` and `Lithology_Group`: Upscales the lithological information to more generic rock groups. For example, `Detailed_Lithology`: "basalt" is upscaled to `Lithology_Subgroup`:"mafic_fine-grained crystalline" and further upscaled to `Lithology_Group`:"igneous rock".

6. Calculated X, Y, Z for the start, mid and endpoint also using the minimum curvature algorithm. The desurveying code was heavily based on the _pyGSLIB_ drill hole module.





**Figure 7.** Lithology extraction is done through the Lithology Code and Comments workflows. The values are fetched from the **dhgeology** and **dhgeologyattr** table (green) using either the Lithology (blue) and Lithology Code (light blue) thesauri or the Comments (blue) thesaurus. The string fetched is then cleaned prior to the fuzzy string matching using the Cleanup Dictionary (dark yellow). The result is then matched against the **Detailed_Lithology** level of the Lithological Hierarchical Thesaurus. If there is a match with a score greater or equal to 80, the match is taken and matched with the rest of the columns in the Lithology Hierarchical thesaurus. If not, it is labelled as unclassified rock.


Once the `Company_Litho` (decoded lithology from `Company_LithoCode`) or from the `Comments` (free text descriptions) have been extracted from the database, the lithology strings were pre-processed such that:

a) The strings were converted to lowercase form.

b) The string inside parenthesis, brackets and braces were removed, as these were found to reduce the accuracy of the matching.

c) The string followed by key phrases such as "with", "possibly", "similar to" were removed.

d) If any of the words listed in the clean-up dictionary were present in the string, these words were removed.

e) Lemmatization, the removal of the inflections at the end of the words in order the "lemma" or root of the words, was applied to all nouns (Müller et al., 2015).

f) All words with non-alphabetic characters and tokens with less than three characters were removed.

g) Stopwords, a set of words frequently used in language which are irrelevant for text mining purposes (Wilbur and Sirotkin, 1992), were removed. Examples on stopwords are: as the, is, at, which, and on.

This is followed by fuzzy string matching, an algorithm which finds the string that matches a pattern approximately. Fuzzy string matching is typically divided into two sub-problems: 1) finding approximate substring matches inside a given string, and 2) finding dictionary strings that match the pattern approximately. Fuzzy string matching uses the Levenshtein Distance to calculate the differences between sequences and patterns (Okuda et al., 1976;Cohen, 2011). The Levenshtein distance measures the minimum number of single-character edits (insertion, deletion, substitution) necessary to convert a given string into an exact match with the dictionary string (Levenshtein, 1965).

We utilized *fuzzywuzzy* (https://github.com/seatgeek/fuzzywuzzy) for this. *fuzzywuzzy* provides two methods to calculate a similarity score between two strings: *ratio()* or *partial_ratio()*. It also provides two functions to pre-process the strings: *token_sort()* and *token_set()*. In this work, we used the *token_set_ratio()* scorer to do fuzzy string matching to classify the database lithology description into one of the lithology thesaurus entries (Table 1). *token_set()* pre-processes the strings by: 1) splitting the string on white-spaces (tokenization), 2) turning to lowercase and 3) removing punctuations, non-alpha non-numeric characters and unicode symbols. It tokenizes both strings (given string and dictionary string), splits the tokens into: intersection and remainder, then sort and compare the strings. Since the sorted intersection component of token *_set()*, will result in an exact match, the score will tend to increase when: 1) the sorted intersection makes up a larger percentage of the full string, and 2) the remainder component are more similar. The *ratio()* then computes the standard Levenshtein distance between two strings. *token_set_ratio()* was found to be effective in addressing harmless misspelling and duplicated words but sensitive enough to calculate lower scores for longer strings (3-10 word labels), inconsistent word order and missing or extra words. *partial_ratio()* which takes the "best partial" of two strings or the best matching on the shorter substring was not preferred as it does not address the difference and order in substring construction. *token_sort()* was not preferred as it alphabetically sorts the tokens that ignores word order and does not weight intersection tokens which does not address the behavior of the strings in the logs.





**Table 1. Examples of fuzzy string matching output using different combinations of the *fuzzywuzzy* functions. The table demonstrates the corresponding effect of these functions to the given string. *Token_set_ratio ()* works best on geological free text descriptions as it weights the intersection tokens, honors substring construction and word order and ignores misspelling, extra and duplicated words. *Partial_ratio ()* ignores substring construction and order and is more sensitive to misspellings. *Token_sort_ratio ()* also ignores substring order and does not recognize duplicate and extra words.**

| *fuzzywuzzy* Function | Given String | Dictionary String | Score | | Remarks |
|---|---|---|---|---|---|
| *ratio ()* | diorite | granodiorite rock | 58 | ✓ | *partial_ratio ()* ignores substring construction |
| *partial_ratio ()* | diorite | granodiorite rock | 100 | ✗ | |
| *ratio ()* | granodoirit rcok | granodiorite rock | 85 | ✓ | *ratio ()* mitigates misspelling |
| *partial_ratio ()* | granodoirit rcok | granodiorite rock | 81 | ✗ | |
| *ratio ()* | rock felsic granodiorite | granodiorite rock | 59 | ✓ | *partial_ratio ()* ignores substring order |
| *partial_ratio ()* | rock felsic granodiorite | granodiorite rock | 83 | ✗ | |
| *token_set_ratio ()* | rock felsic granodiorite | granodiorite rock | 83 | ✓ | *token_sort_ratio ()* ignores substring order |
| *token_sort_ratio ()* | rock felsic granodiorite | granodiorite rock | 100 | ✗ | |
| *token_set_ratio ()* | intermediate granodiorite rock | granodiorite rock | 100 | ✓ | *token_set_ratio ()* weights intersection tokens |
| *token_sort_ratio ()* | intermediate granodiorite rock | granodiorite rock | 72 | ✗ | |
| *token_set_ratio ()* | gray granodiorite granodiorite | granodiorite rock | 83 | ✓ | *token_set_ratio ()* ignores extra and duplicate words |
| *token_sort_ratio ()* | gray granodiorite granodiorite | granodiorite rock | 64 | ✗ | |
| **token_set_ratio ()** | **gray granodiorite granodiorite rckso** | **granodiorite rock** | **83** | ✓ | **token_set_ratio () weights intersection tokens, addresses substring construction and word order, ignores misspelling, extra and duplicate words** |
| *partial_token_set_ratio ()* | gray granodiorite granodiorite rckso | granodiorite rock | 100 | ✗ | |

The code calculates the to*ken_set_ratio()* between the `Company_Litho` or `Comments` (given string) and the entries in the lithology hierarchical thesaurus (dictionary string). The tendency of geologists when describing rocks is to enumerate the descriptors before the rock name. For example, if the lithology in the logged interval is "basalt", the free text description could be something like "Dark gray to dark reddish brown, with olivine phenocrysts, largely altered andesitic basalt". After processing the string, it will be left with "andesitic basalt". To avoid, misclassifying the rock to "andesite", a bonus score is also added to add weight to the last word (in this case, "basalt") (Appendix C2 Pseudocode). For the pair between `Company_Litho` or `Comments` and the entries in the lithology hierarchical thesaurus with the highest score, the first synonym is stored as `Detailed_Lithology`. If the score is less than 80, it is classified as "unclassified rock". The cut-off value of 80 is user-defined, and in this case chosen based on the performance of the matching on a subset of 1,548 unique lithology codes (Fig. 8) from a subset of the YSGB dataset in the Golden Grove area. The matching performance may vary depending on the dataset being extracted. It is advised to test in a subset and adjust these cut-off score depending on these results. If the performance is significantly lower, this indicates that the thesauri used in *dh2loop* may not be suitable to your area. The user may opt to update these thesauri to suit their needs. Once matched on `Detailed_Lithology`, the corresponding `Lithology_Subgroup` and `Lithology_Group` classifications are also fetched.

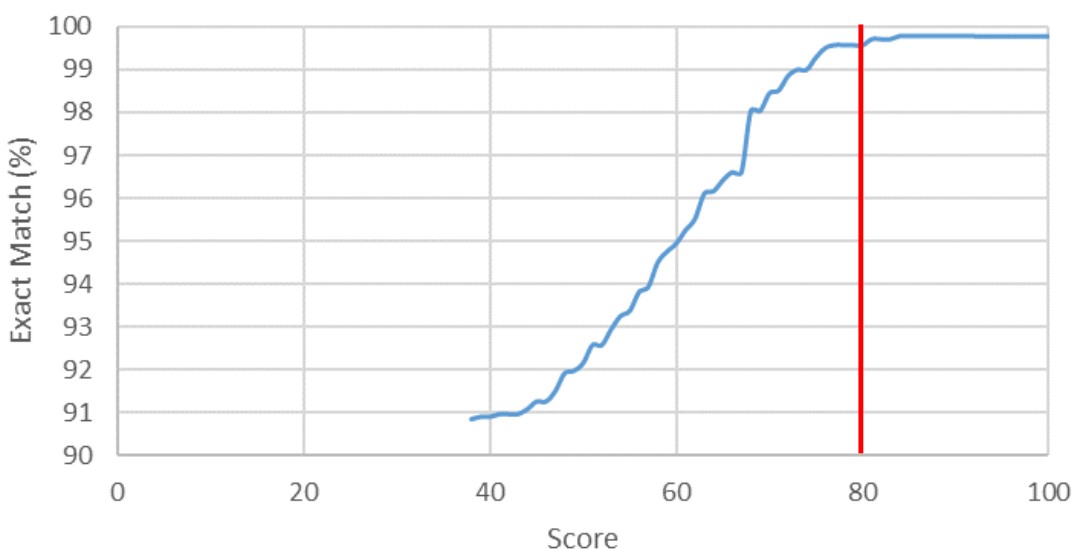


**Figure 8. The user-defined cut-off score of 80 was chosen based on the results of the testing different cut-offs on a smaller dataset within the YSGB area. As seen in this figure, the number of exact matches plateau at a score of 80. This relationship may vary depending on the datasets available in the area. Thus, this cut-off value is user-defined and is best to test the matching performance on a subset in the user's area.**

## 3 Data Extraction Results

### 3.1 Collar

Extraction of the collar data for YSGB resulted in a collar file with 68,729 drill holes (Table 2). This information was extracted from the `collar` table with 73,881 drill holes with 769,981 rows of information from `collarattr`. It includes the location of the collar both in geographic and projected coordinated systems, relative level (`RL`) and maximum depth (`MaxDepth`). A

total of 136,100 records for `RL` were retrieved from the database, 1,526 of which were disregarded: 846 records for having an `RL` value greater than 10,000 meters and 680 non-numeric records. These discarded values were retrieved from the attribute column "RL_Local". In spite of it being an isolated issue for "RL_Local", the attribute column was retained as it is retrieved sensible values for other companies. The discarded values were limited to data from two companies (4085, 4670) for `RL` attribute columns "TD" and "DEPTH". A total of 58,706 records for `MaxDepth` were retrieved from the database: 58,642 of

which were extracted as is, while 64 entries were disregarded for having a value of -999. The discarded values come from 8 companies. Null values are assigned to disregarded and absent `RL` or `MaxDepth` values. The "clean" collar export file contains at least either a value for `RL` or `MaxDepth`. The reasoning behind keeping records with at least one of the two field is there are other ways to extract for `RL` or `MaxDepth` from the database. `RL` values can be extracted from digital terrain models and `MaxDepth` values can be taken for the largest `ToDepth` values from the other tables.

### 3.2 Survey

For the survey extraction, the `dhsurvey` table contained 146,713 survey depth intervals (from 45,708 drill holes) with corresponding 850,507 entries of supplementary survey information in `dhsurveyattr` (Table 3). Survey extraction in YSGB resulted in 126,669 survey depth information across 45,708 drill holes with azimuth (-52.5 to 359) and dip measurements (0-90) for each depth interval. A total of 517,592 records for `Azimuth` were retrieved from the database. 77



`Azimuth` values greater than 360 were retrieved and thus disregarded. 152 values were non-numeric values and were also disregarded.  These discarded values involved 228 holes across 10 companies. A value of 0 was assigned to missing `Azimuth` values. A total of 118,223 records for `Dip` were fetched from the database, 118,138 of which were extracted as is, while 95 entries were disregarded for having a value greater than 90. A values of -90 was assigned as the default for `Dip`. The discarded values correspond to 94 drill holes across 5 companies.

**3.3 Lithology: Lithology Code and Comments workflow**

Lithology extraction is divided into two workflows. For the Lithology Code workflow, the extraction starts with filtering the *dhgeology* and *dhgeologyattr* table by the location extents and the Lithology thesaurus. The *dhgeology* table contained 47,062 drill holes across 115 companies with 797,975 lithology depth intervals with corresponding 820,612 entries of lithology information in *dhgeologyattr*. These records were matched with the entries from the Lithology Code

thesaurus resulting to 273,684 matched records. The `FromDepth` and `ToDepth` for these records were then validated. 74 records had equal `FromDepth` and `ToDepth` values. 654 had values for `FromDepth` but null values for `ToDepth`. For both cases, `ToDepth` was calculated as `FromDepth`+0.01. The Lithology Code workflow resulted to 273,684 intervals across 12,793 drill holes wherein 235,606 records were successfully matched in the fuzzy string matching.

The Comments workflow extracts the records from the *dhgeology* and *dhgeologyattr* table as well, but this time using the Comments Thesaurus. For YSGB, the database has 262,567 records across 22,766 drill holes with comments. Since the comments are extracted here to compare their results from fuzzy string matching, only those records that matched in the Lithology Code workflow were retained. This resulted to 47,823 records, however, only 7,870 records were successfully matched on `Comments`. The dataset for the fuzzy string matching assessment (Sect. 5) consists only of the unique records

matched on both workflows (3,074 records). It was visually checked from the records that the Lithology Code `Detailed_Lithology` results were sound classifications of the `Company_Litholog`y. This was done to make sure that these results could be considered as the "true value" in the fuzzy string matching assessment (Sect. 5).





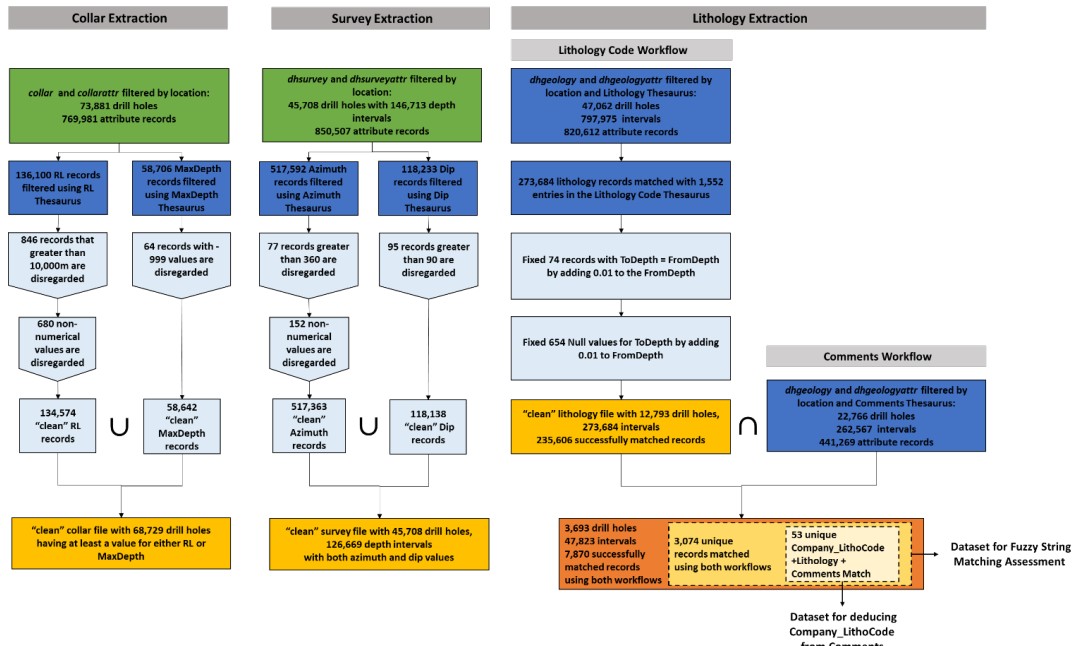

**Figure 9. Extraction of the collar, survey and lithology data for the YSGB. The collar extraction resulted in a collar file with 68,729 drill holes from the *collar* table with 73,881 drill holes with 769,981 rows of information from *collarattr*. A total of 136,100 records for RL were retrieved from the database, 1,526 of which were disregarded: 846 records for having an RL value greater than 10,000 meters and 680 non-numeric records. A total of 58,706 records for MaxDepth were retrieved from the database: 58,642 of which were extracted as is, while 64 entries were disregarded for having a value of -999. The "clean" collar export file contains at least either a value for RL or MaxDepth. Survey extraction in YSGB resulted in 126,669 survey depth information across 45,708 drill holes. The *dhsurvey* table contained 146,713 survey depth intervals (from 45,708 drill holes) with corresponding 850,507 entries of supplementary survey information in *dhsurveyattr*. 77 Azimuth values greater than 360 and 152 values were non-numeric values. Lithology extraction is divided into two workflows. For the Lithology Code workflow, the extraction starts with filtering the *dhgeology* and *dhgeologyattr* table by the location extents and the Lithology thesaurus. The *dhgeology* table contained 47,062 drill holes across 115 companies with 797,975 lithology depth intervals with corresponding 820,612 entries of lithology information in *dhgeologyattr*. These records were matched with the entries from the Lithology Code thesaurus resulting to 273,684 matched records. The FromDepth and ToDepth for these records were then validated. 74 records had equal FromDepth and ToDepth values. 654 had values for FromDepth but null values for ToDepth. For both cases, ToDepth was calculated as FromDepth+0.01. The Lithology Code workflow resulted to 273,684 intervals across 12,793 drill holes wherein 235,606 records were successfully matched in the fuzzy string matching. The Comments workflow extracts the records from the *dhgeology* and *dhgeologyattr* table as well, but this time using the Comments Thesaurus (262,567 records across 22,766 drill holes with comments). 47,823 records were present in both workflows, 7,870 records of which were successfully matched. The 3,074 unique entries from this was used as the dataset for the fuzzy string matching assessment (Sect. 5).**


**4    Unique Lithology Code Results**

Using the `Company_LithoCode`, `Lithology_Code Detailed_Lithology` and `Comments Detailed_Lithology` from the dataset for the fuzzy string matching assessment, we can assess if matches using the `Comments` workflow alone can sufficiently decode lithology. Excluding the unmatched entries and taking only the unique combinations of `Company_LithoCode`, `Lithology_Code Detailed_Lithology` and `Comments` 540 `Detailed_Lithology`, the dataset results into 53 unique records.

To be able to assess the matching we take a look at the type of matches between `Lithology_Code Detailed Lithology` and `Comments Detailed_Lithology`. First, we define a **match** as retrieving an answer from the fuzzy string matching with a score greater than 80. It is important to note here that it only suggests that it succeeded to find an answer 545 above the score threshold but not necessarily mean that it is the correct answer. To further describe the quality of a match, we modified for this purpose the following terminologies from the   Simple Knowledge Organization System (Miles and Bechhofer, 2009):

 a) **Exact Match** suggests that both workflows resulted in the same classification at all 3 levels. The match at the `Detailed_Lithology` level has an exact match, thus resulting to an exact match on the other two levels.

b) **Close Match** suggests that the results at the `Detailed_Lithology` level are related rocks and belong to the same `Lithology_Subgroup`. This is usually caused by differing use of lithological nomenclature.

 c) **Related Match** suggests that the results at the `Detailed_Lithology` level are related rocks and belong to the same `Lithology_Group`.

 d) **Broad Match** refers to the `Detailed_Lithology` from Lithology Code workflow matches to a 555 `Lithology_Subgroup` in the Comments workflow.

 e) **Narrow Match** is the logical equivalent of a Broad Match. In this case, the Comments workflow resulted in a `Detailed_Lithology` level while the Lithology Code workflow resulted in a `Lithology_Subgroup` level.

 f) **Broader Match** is similar to a broad match except that the `Detailed_Lithology` from Lithology Code workflow matches to a `Lithology_Group` instead of a `Lithology_Subgroup` in the Comments workflow.

g) **Narrower Match** is the logical equivalent of Broader Match. The Comments workflow results to a `Detailed_Lithology` while the `Lithology_Code` workflow results to a `Lithology_Group` level.

 h) **Failed Match** suggests all levels of both workflows do not match. This is usually attributed to contrasting information from both fields or the algorithm fails. This category is an addition to the SKOS reference.

For better understanding of these relationships, examples are shown in Table 2 and Fig 9.





**Table 2.** Fuzzy string matching terminology used to describe the quality of matches based on the Simple Knowledge Organization System (SKOS) (Miles and Bechhofer, 2009). The values being compared are the `Detailed_Lithology` level for both Lithology
Code and Comments workflow (brown text). The level at which the records are considered to match are in bold. A Match retrieves an answer from the fuzzy string matching with a score greater than 80. An Exact Match suggests that both workflows resulted in the same classification at all 3 levels. A Close Match suggests that the results at the `Detailed_Lithology` level are related rocks and belong to the same `Lithology_Subgroup`. A Related Match suggests that the results at the `Detailed_Lithology` level are related rocks and belong to the same `Lithology_Group`. A Broad Match refers to the `Detailed_Lithology` from
Lithology Code workflow matches to a `Lithology_Subgroup` in the Comments workflow. Narrow Match is the logical equivalent of a Broad Match. Broader Match is similar to a broad match except that the `Detailed_Lithology` from Lithology Code workflow matches to a `Lithology_Group` instead of a `Lithology_Subgroup` in the Comments workflow. Narrower Match is the logical equivalent of Broader Match. A Failed Match suggests all levels of both workflows do not match.

| Lithology Code Detailed Lithology Level | Comments Detailed Lithology Level | Lithology Code Lithology Subgroup Level | Comments Lithology Subgroup Level | Lithology Code Group Level | Comments Group Level | Type of Match |
|---|---|---|---|---|---|---|
| **basalt** | **basalt** | | | | | Exact Match |
| basalt | basic volcanic rock | **mafic fine grained crystalline** | **mafic fine grained crystalline** | | | Close Match |
| basalt | gabbro | mafic fine grained crystalline | mafic coarse grained crystalline | **igneous** | **igneous** | Related Match |
| basalt | mafic fine grained crystalline | **mafic fine grained crystalline** | **mafic fine grained crystalline** | | | Broad Match |
| mafic fine grained crystalline | basalt | **mafic fine grained crystalline** | **mafic fine grained crystalline** | | | Narrow Match |
| basalt | mafic | mafic fine grained crystalline | mafic | **igneous** | **igneous** | Broader Match |
| mafic | basalt | mafic | mafic fine grained crystalline | **igneous** | **igneous** | Narrower Match |
| basalt | sandstone | mafic fine grained crystalline | clastic | igneous | sedimentary | Failed Match |




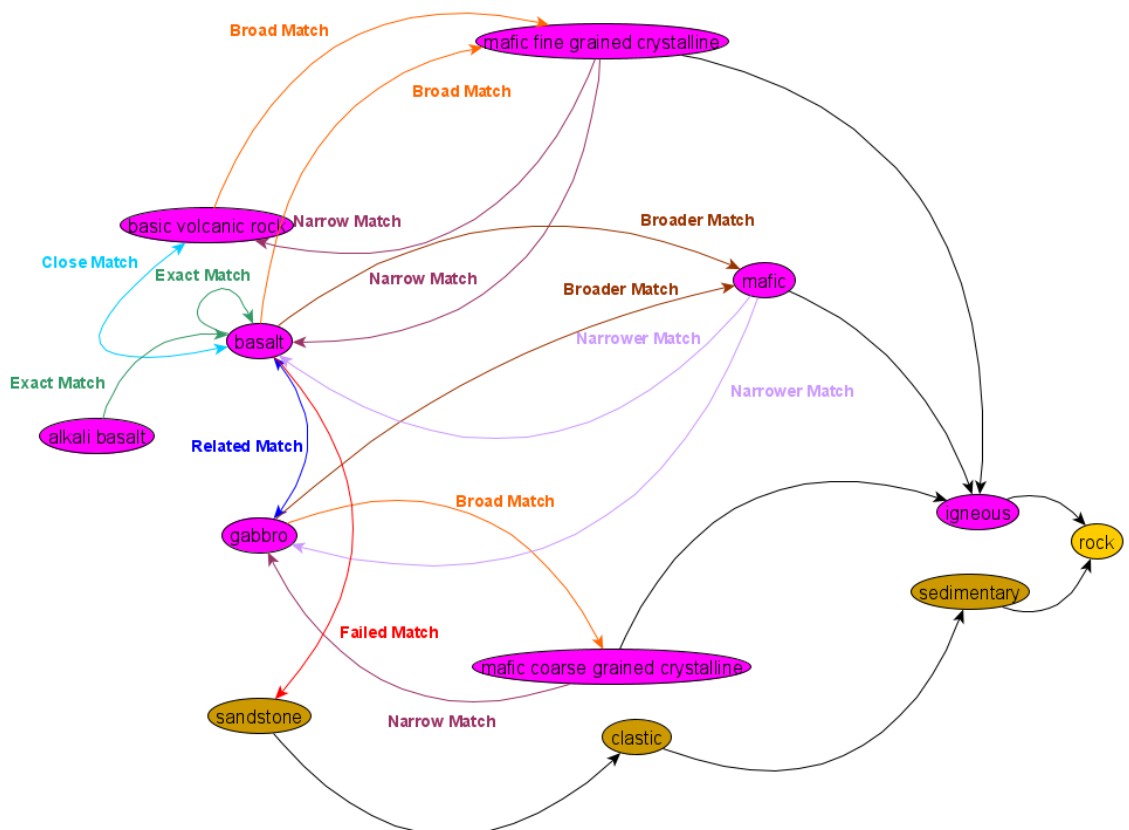

**Figure 10. SKOS graph showing the semantic, associative and hierarchical relationship in the Lithology Hierarchical Thesaurus. In this example, terms "basalt" and "alkali basalt" are judged to be sufficiently the same to assert an Exact Match relationship (in green). "basic volcanic rock" however is considered a Close Match (in cyan) and "gabbro" a Related Match (in blue). "mafic fine grained crystalline" and "mafic coarse grained crystalline" are broader concepts, thus considered a Broad Match (in orange) to**

**"basalt" and "gabbro" respectively. Broader Match (in brown) are similar to Broad Matches but are used to refer a wider semantic difference between the two concepts. Narrow Matches (in light purple) and Narrower Matches (in dark purple) are the logical equivalent of Broad Match and Broader Match. Failed Matches is used to describe unrelated matches.**

out of the 53 unique entries (64%) result to matches between the `Detailed Lithology` and `Comments Detailed_Lithology`. 26 of which are Exact Matches, 19 unique entries are Close Matches and 26% percent are Failed

Matches. The Failed Matches are due to unrelated descriptions in the `Comments` field which was used to obtain the results in `Comments Detailed_Lithology`. An example of this is the interval is logged as "ironstone" but the comments contain "mafic schist". Another less common reason is the `Company_LithoCode` is repeated in the `Comments`. An example of this is would be an interval logged as "colluvium" and the `Comments` as "COL". The Comments workflow will result to "coal" instead.

**5   Fuzzy String Matching Results**

We present results from the data extraction using both workflows: Lithology Code and Comments. The overlaps between these two workflows suggest that the user may need to make choices to identify which is better suited for matching in their area of interest. To better understand the difference between these results, we take a looks at the matching overlaps between the two workflows (3,074 entries). These matching overlaps are used to compare and describe the fuzzy string matching using the

decoding the lithology codes and using the comments free text descriptions.





**Exact Matches**: Of the total matched entries, 944 were Exact Matches (31%) (Table ). The Exact Matches are ideal outcomes as both workflows resulted in exactly the same answers.

**Close Matches**: The Close Matches are common for coarse-grained igneous rocks, clastic sedimentary rocks, surficial residual rocks and filling structures. The coarse-grained igneous rocks such as gabbro, gabbroid and dolerites are used interchangeably in both fields. `Comments` can contain terminologies such as "gabbroic", "granophyric gabbro to dolerite", "intrusive granitoid to gabbro" resulting to close matches. Similar cases are observed between granodiorite and granite and between peridotite and coarse-grained ultramafic rocks. For clastic sedimentary rocks, the Close Matches are a result of gradation of grain size in the 610 comments field. For example, an interval logged as mudstone is then described in the comments as "mudstone to sandstone" or "intercalated with siltstone". These comments will result in "sandstone" and "siltstone", respectively. Both clastic sedimentary rocks but not an Exact Match to mudstone. Metasediments and quartz veins occur together and what is described last dictates the `Detailed_Lithology` classification. Surficial rocks such as soil, duricrust, colluvium, laterite, calcrete, ferricrete and cover are used loosely or occur together resulting to multiple combination of these Close Matches.


**Related Matches**: 60 entries (3%) resulted in related matches. For igneous rocks, this result is observed when the comments field use rock type descriptors such as "komatitic", "basaltic" and "doleritic". An example would be an interval logged as dolerite and is then described in the comments as "dolertic basalt". This would result in dolerite in the Lithology Code workflow and "basalt" in the Comments workflow. Both lithologies are igneous, however have different composition and 620 textural implications. For sedimentary rocks, Lithology Code workflow results to sedimentary rocks classified based on grain size as they have been logged ("gravel", "mud"). The comments field contains compositional descriptions such as "with silcrete" or "minor chert". In this case, the comments workflow will result in "silcrete" and "chert". Both workflows will result in sedimentary rocks, but the Lithology Code workflow will result in "clastic" rocks while the comments workflow will classify these to "siliceous" at the `Lithology_Subgroup` level. The related matches for structures occur across coincident 625 lithologies such as "mylonite", "vein", "fault" and "breccia" which could either be "fillings" or "fault_rock" at the `Lithology_Sugbroup`.

**Broad and Narrow Matches**: No broad matches were noted and only one narrow match was obtained (Table 3). The interval was logged as "ironstone" with "BIF" in the comments, "ironstone" being a more general description for "banded iron 630 formation".

**Broader and Narrower Matches**: More common cases are Broader and Narrower Matches indicate that there is a bigger relationship gap between the data in the lithology and comments field. Broad matches are a result of low detail comments. For example, an interval logged as "gabbro" is described as "medium-grained mafic", "massive mafic", "rich mafic". The inverse 635 is noted for narrower matches, the interval is logged as "sediment" but in the comments the interval is described as "siliceous sediments".

**Failed Matches:** 1,694 entries resulted in Failed Matches (55%). Failed Matches occur when the lithology and comments field contain different information. This could be because of the lithology contains the main lithology while the comments contains all other lithologies intercalated in the interval. Another reason is the lithology field is relogged based on adjacent 640 intervals without amending the comments. "Mudstone" had failed matches with a wide range of lithologies, such as: "amphibolite", "dolerite", "saprolite", "duricrust", "laterite", "banded iron formation", "chert", "phyllite", "schist", "vein".





The same is observed for igneous rocks such as: "coarse-grained-ultramafic-rock". For "chert", the failed matches are within a range of sedimentary rocks: "alluvium" and "mud", "amphibolite" and "massive sulphide", "carbonate", "vein", "pegmatite".


**Table 3. Distribution of matches across the Fuzzy String Matching Dataset. A total of 45% of the unique records were matched reasonably, 31% of which are Exact Matches, 6% Close Matches, 3% Related Matches, 3% Broader Matches and 3% Narrower Matches.**

| Type of Match | Number of Entries | Percent |
|---|---|---|
| Exact Match | 944 | 31% |
| Close Match | 197 | 6% |
| Related Match | 60 | 3% |
| Broad Match | 0 | 0% |
| Narrow Match | 1 | 0% |
| Broader Match | 84 | 3% |
| Narrower Match | 95 | 3% |
| Failed Match | 1694 | 55% |
| **TOTAL** | **3,074** | **100%** |

The matching results can be visualized as confusion matrices, which are typically used in machine learning to compare the performance of an algorithm versus a known result. In this case, we are comparing the performance of the string matching using the Comments workflow against the results from the Lithology Code workflow. From the 3,074 unique records, we use a total of 1,200 samples for the confusion matrices. The reason for this difference is the limitation of building a confusion matrix wherein both workflows look at the same classes. Each row of the matrix represents the matched lithology from the

Comments workflow while each column represents the matched lithology from the Lithology Code workflow. The diagonal elements represent the count for which the Comments workflow class is equal to the Lithology Code workflow. The off-diagonal elements are those that are misclassified by the Comments workflow. The higher the diagonal values of the confusion matrix the better, indicating many correct matches. The confusion matrices show normalisation by class support size. This kind of normalisation addresses the class imbalance and allow better visual interpretation of which class is being misclassified.

The color of the cell represents the normalised count of the records to address the uneven distribution of records across different classes.

Relying on one metric to assess the matching can be misleading, therefore, we would like to use a couple of metrics: accuracy, precision, recall and F1 score. Accuracy sums the true positives and true negatives and puts this number in the contrast of all

matches:

$$Accuracy = \frac{True\ Positive + True\ Negative}{Positive + Negative}$$

Precision is a useful metric in cases where false positives are a higher concern than false negatives. The precision of the matching is the true positives divided by the sum of the true positives and false positives. The precision measures the fraction

of correctly classified are positive:

$$Precision = \frac{True\ Positive}{True\ Positive + False\ Positive}$$

Recall is a useful metric in cases where false negatives trumps false positives. It is the percentage of total relevant results correctly classified while precision is the percentage relevant results. It is computed as:





$$Recall = \frac{True\ Positive}{True\ Positive + False\ Negative}$$

F1-score is a combined metric of precision and recall. It takes their harmonic mean, thus it is maximum when precision is equal to recall. However, the interpretability of the F1-score is poor. Its formula is written as:

$$Matching\ F1\ Score = \frac{2}{\frac{1}{Recall} + \frac{1}{Precision}}$$

### 5.1 Structure and Texture

While geological structures are not lithologies, they are sometimes described in lithological logs (Fig 11). Structures common
in the YSGB area are faults and veins. Figure 11 shows the confusion matrix for the structures and textures. The vertical axis represents the matches from the Lithology Code workflow while the horizontal axis for the results from the Comments workflow. We consider a dataset of 52 unique records where we are trying to assess if the Comments workflow results to the same classification as the Lithology Code workflow. Figure 11 shows that there are 6 records classified as "fault" and 46 records as "vein". When looking at the classification of "faults" we can say that there are 2 records that are true positives. 46
records are true negative pairs, as in this 2x2 matrix, if it is not a "fault", it is a "vein". True negatives together with true positives are the Exact Matches and suggests that the Comments workflow identified it correctly. To have a better look at the parts that were not classified correctly we look at the false positives and false negatives. False positives represent the number of records classified as "fault" but based on the Lithology Code workflow are not. In this case, there are no false positive values. False negatives represent number of records classified as "vein" but are actually "faults" based on the Lithology Code
workflow.

A total of 48 Exact Matches were noted, 46 records of which are "veins" and 2 records are "faults". This can be surmised by looking into the diagonal cells. The rest of the "veins" (4 records) are Related Matches as "faults". They are considered Related Matches as faults and veins tend to coexist in nature. In addition, faults often occur as fault zones, with infill clay or silica vein
sulphides which are described in the comments that then obscures the classification. These structure-related lithological descriptions can be used as proxies in further geological studies.



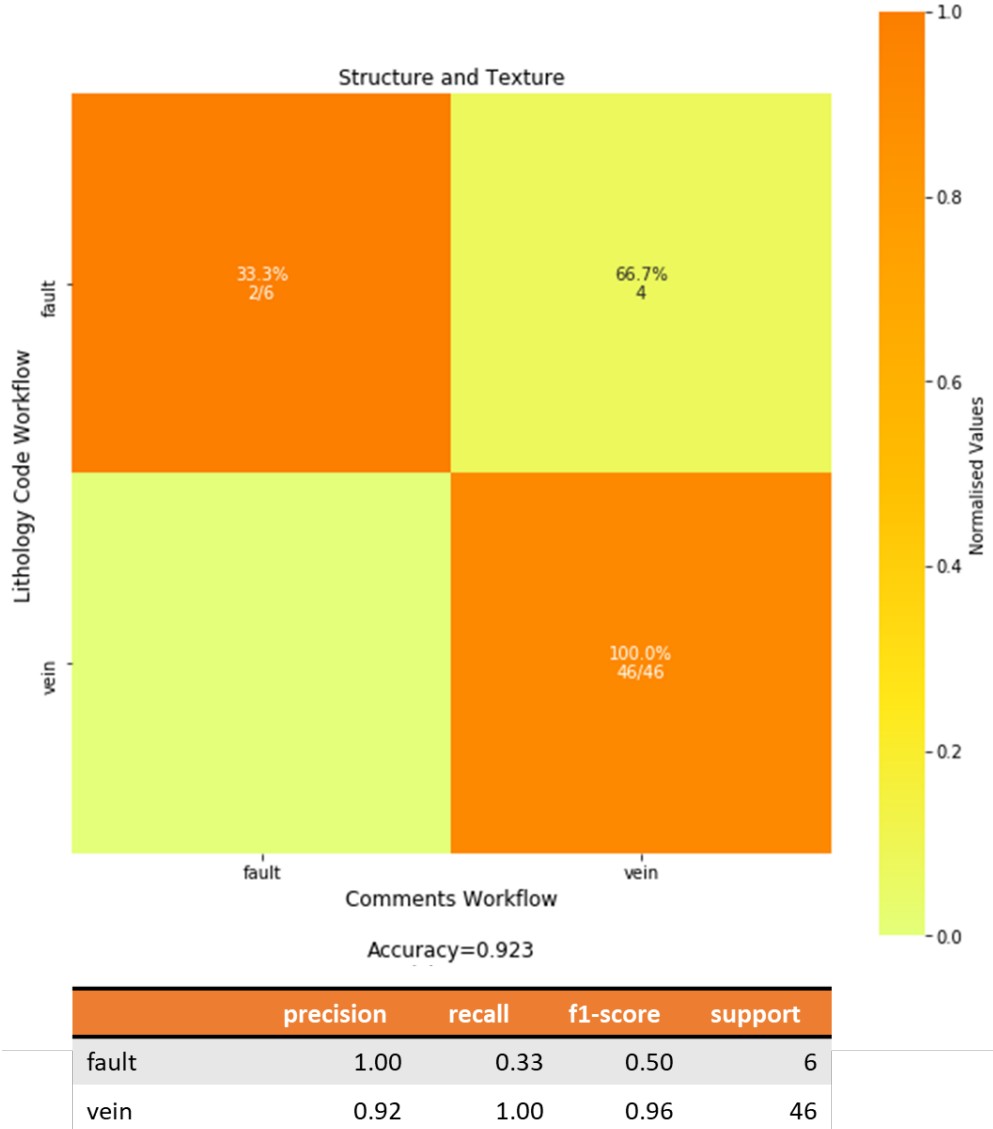

| | precision | recall | f1-score | support |
|---|---|---|---|---|
| fault | 1.00 | 0.33 | 0.50 | 6 |
| vein | 0.92 | 1.00 | 0.96 | 46 |
| **macro avg** | **0.96** | **0.67** | **0.73** | **52** |
| **weighted avg** | **0.93** | **0.92** | **0.91** | **52** |

**Figure 11. Confusion matrix for structure and texture comparing the fuzzy string matching results from the Lithology_Code workflow (vertical axis) and Comments workflow (horizontal axis). The heatmap shows the values normalised to the support size to address the imbalance between classes. The values shown in the cells indicate the number of samples classified for the class. Empty cells indicate zero samples. The Structures and Texture `Lithology_Group` had an accuracy of 92.3% across 52 samples, 46 for veins and 6 for faults.**

### 5.2 Igneous Rocks

The confusion matrix for igneous rocks considers a dataset of 218 unique records (Fig 12). Dealing with a larger matrix is not

as straight-forward as the previous matrix. When looking at the classification of a single lithology, the true positives are where

both axes refer to the same class. For example, for "basalt" there are 15 records of true positives which correspond to the Exact



Matches. The false positives are the sum of all the other entries along the corresponding vertical axis and the false negatives are the sum of all the entries along the corresponding horizontal axis. The sum of all the other cells represent the true negatives. For "basalt", there are 15 true positives, 13, false positives, 15 false negatives and 175 true negatives. This results to 54%

classification precision for "basalt".

This statistic is helpful in quantifying the performance of the classification. However, what it does not capture is the semantic and hierarchical relationship of the false negative pairs. As shown in Figure 12, 3 records were classified as "komatiite" and 12 records were classified as "mafic". The "komatiite" matches are a result of when the comments describe the basalts as

"komatiitic basalts". This can be considered as a Related Match. The 12 records which were classified as "mafic: are considered "Broader Match". For the false positive values, the "mafic" records are Narrower Matches while the "dolerite" is a Related Match. These quantitative assessment of the matches show us that although the matching is not perfect, the context of the misclassification is not severe.

"Dolerite" is the most common igneous rock matched. This could be attributed to the sampling bias towards dolerite as it is often targeted by drilling as they are used as targeting criteria for gold mineralisation (Groves et al., 2000). Given that dolerites can be described by their mafic component or be confused as gabbro when weathered, the descriptions contain strings "mafic" and "gabbro" which explain Close and Broader Matches. Gabbros are also common in the YSGB. Some of the "gabbros" were classified as "mafic" in the `Comments Detailed_Lithology`. This is another example of a Broader Match. However,

it is important to note that although it is not an Exact Match, a Broader Match can be useful in geological studies relating to rock composition as gabbros are members of mafic rocks. 40% of the igneous rock that were mismatched at the `Detailed_Lithology` level were Broader Matches (matches correctly at `Lithology_Group`).



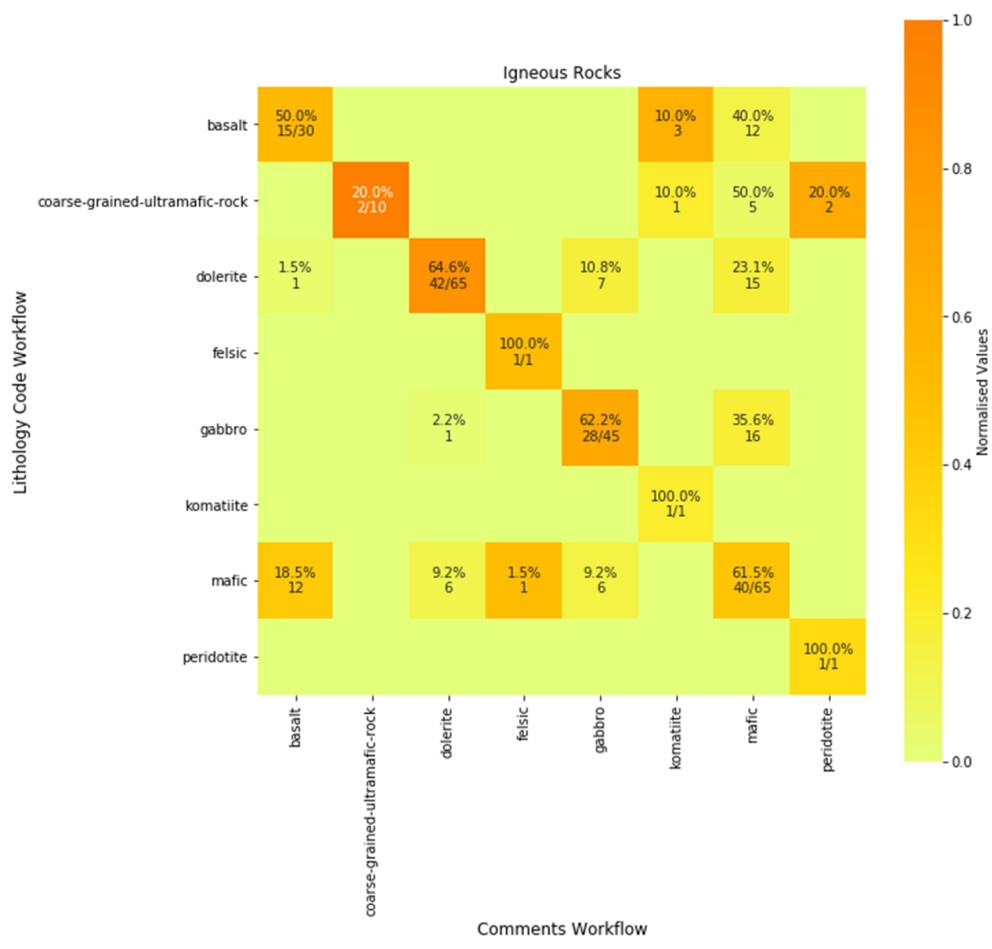

| | precision | recall | f1-score | support |
|---|---|---|---|---|
| basalt | 0.54 | 0.50 | 0.52 | 30 |
| coarse-grained-ultramafic-rock | 1.00 | 0.20 | 0.33 | 10 |
| dolerite | 0.86 | 0.65 | 0.74 | 65 |
| felsic | 0.50 | 1.00 | 0.67 | 1 |
| gabbro | 0.68 | 0.62 | 0.65 | 45 |
| komatiite | 0.20 | 1.00 | 0.33 | 1 |
| mafic | 0.45 | 0.62 | 0.52 | 65 |
| peridotite | 0.33 | 1.00 | 0.50 | 1 |
| macro avg | 0.57 | 0.70 | 0.53 | 218 |
| weighted avg | 0.66 | 0.60 | 0.60 | 218 |

**Figure 12. Confusion matrix for igneous rocks comparing the fuzzy string matching results from the Lithology_Code workflow**
**(vertical axis) and Comments workflow (horizontal axis). The heatmap shows the values normalised to the support size to address the imbalance between classes. The values shown in the cells indicate the number of samples classified for the class. Empty cells indicate zero samples. The accuracy is 59.6%, with a weighted average precision of 66% and recall of 60%. These results were taken**





**from a subset of 218 samples across 8 classes. "Coarse-grained-ultramafic-rock" has a precision of 1 that implies there are no False Positives.**

**5.3 Sedimentary Rocks**

The largest `Lithology_Group` of the lithological entries relates to sedimentary rocks (800 entries) (Fig 13). 457 of the 800 entries are true positive classification of mudstones. Mudstones are common as shale beds. Mudstones resulted in Related Matches with "chert" and "ironstone". The misclassification occurs when the logs describe intervals wherein the mudstone occurs together and is intercalated with these lithologies. A few mudstones (17) are matched as sandstone due to textural and

grain-size descriptors (Close Match). 48% of the cherts are resulted in Exact Matches. 39 records of cherts resulted in Failed Matches as their `Detailed_Lithogy` level matched with "banded iron formation", it occurs when intercalated together such as "cherts with BIF" or as include string descriptors such as "BIF-fy".





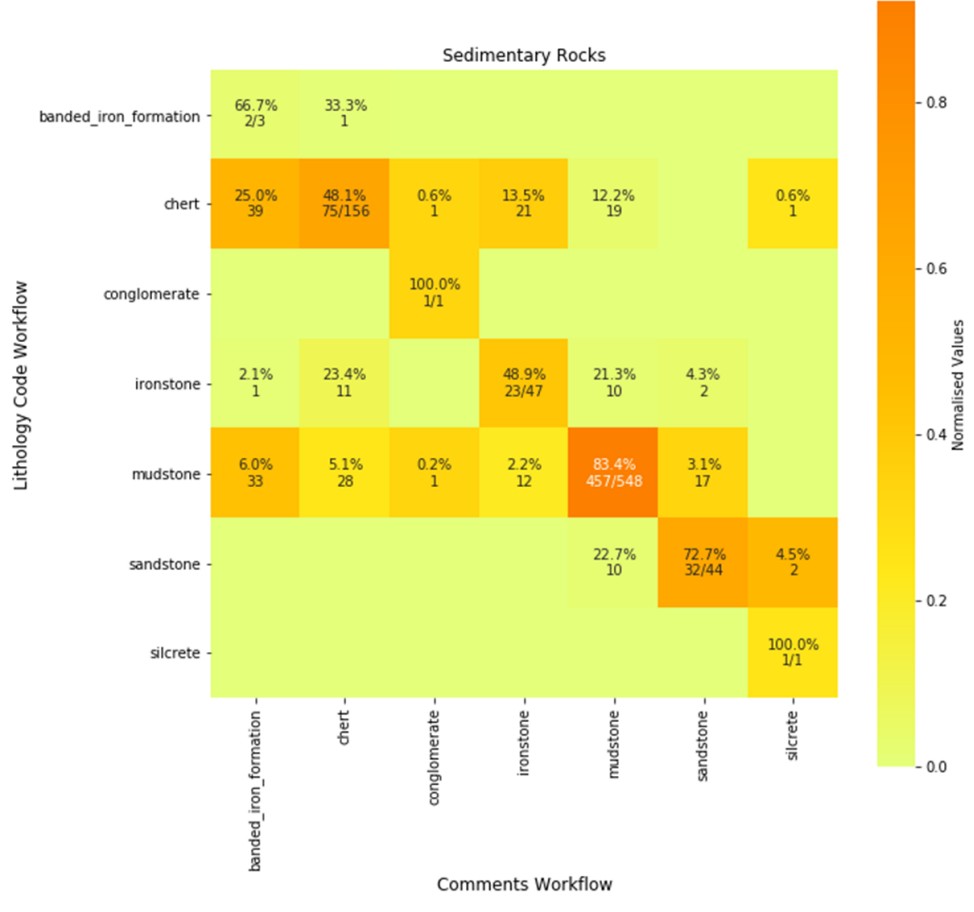

| | precision | recall | f1-score | support |
|---|---|---|---|---|
| banded_iron_formation | 0.03 | 0.67 | 0.05 | 3 |
| chert | 0.65 | 0.48 | 0.55 | 156 |
| conglomerate | 0.33 | 1.00 | 0.50 | 1 |
| ironstone | 0.41 | 0.49 | 0.45 | 47 |
| mudstone | 0.92 | 0.83 | 0.88 | 548 |
| sandstone | 0.63 | 0.73 | 0.67 | 44 |
| silcrete | 0.25 | 1.00 | 0.40 | 1 |
| macro avg | 0.46 | 0.74 | 0.50 | 800 |
| weighted avg | 0.82 | 0.74 | 0.77 | 800 |

**Figure 13. Confusion matrix for sedimentary rocks comparing the fuzzy string matching results from the Lithology_Code workflow (vertical axis) and Comments workflow (horizontal axis). The heatmap shows the values normalised to the support size to address the imbalance between classes. The values shown in the cells indicate the number of samples classified for the class. Empty cells indicate zero samples. The accuracy is 73.9%, with a weighted average precision of 82% and recall of 74%. These results were taken from a subset of 800 samples across 7 classes.**



### 5.4 Metamorphic Rocks

Out of a total of 61 metamorphic rock entries, 60 were matched correctly (Fig 14). Most of these were "schists" as the YSGB area is rich in talc-carbonate schists. The `Company_Lithology` entry "amphibolite mica schist" which was matched as "amphibolite" matches as "schist" in the Comments workflow. This is considered a Related Match.

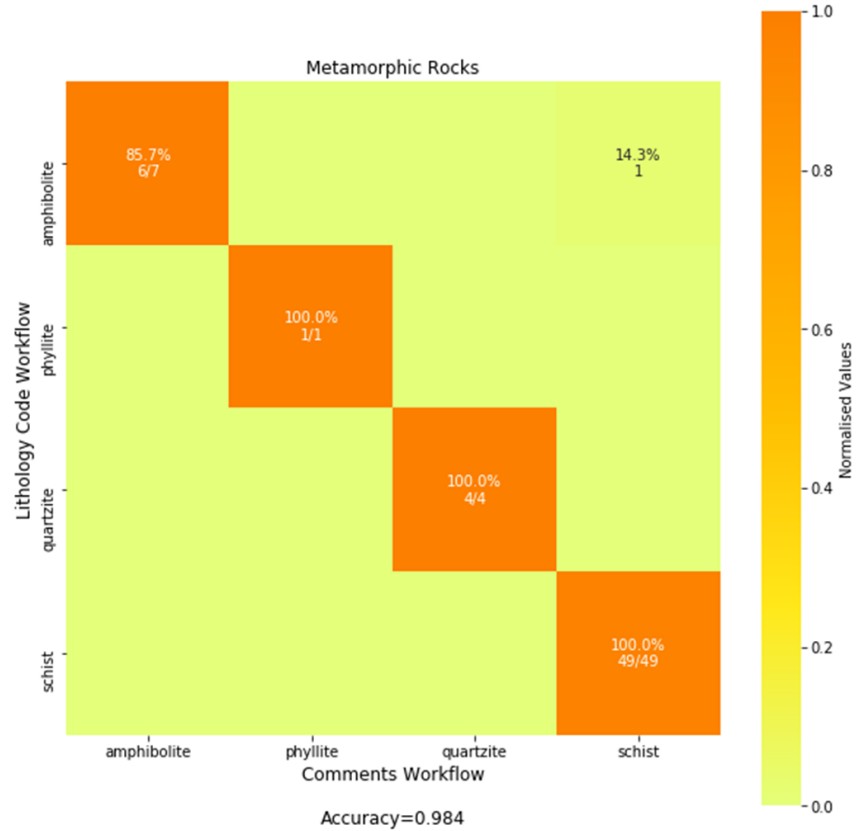

| | precision | recall | f1-score | support |
|---|---|---|---|---|
| **amphibolite** | 1.00 | 0.86 | 0.92 | 7 |
| **phyllite** | 1.00 | 1.00 | 1.00 | 1 |
| **quartzite** | 1.00 | 1.00 | 1.00 | 4 |
| **schist** | 0.98 | 1.00 | 0.99 | 49 |
| **macro avg** | 0.99 | 0.96 | 0.98 | 61 |
| **weighted avg** | 0.98 | 0.98 | 0.98 | 61 |

**Figure 14. Confusion matrix for metamorphic rocks comparing the fuzzy string matching results from the Lithology_Code workflow (vertical axis) and Comments workflow (horizontal axis). The heatmap shows the values normalised to the support size to address the imbalance between classes. The values shown in the cells indicate the number of samples classified for the class. Empty cells indicate zero samples. The accuracy is 98.4%, with a weighted average precision of 98% and recall of 98%. These results were taken from a subset of 61 samples across 4 classes.**



### 5.5 Surficial Rocks

Fuzzy string matching accuracy of surficial rocks scored a 45% on a total of 69 entries (Fig 15). Saprolites were matched as saprolite (Exact Match), rock (Failed Match) and saprock (Close Match). In instances where saprock was inputted as "sap rock", it results to a failed match as "rock". "Soil" is commonly used in logs to refer to the first intercept of highly weathered, clay-rich and unidentifiable intercept. "Soil" was classified with the highest variability of terms: "soil" (Exact Match), "rock" (Failed Match), "duricrust" (Close Match), "colluvium" (Related Match) and "calcrete" (Close Match). "Laterite" was matched to "colluvium" (Related Match), "duricrust" (Close Match) and "lag" (Close Match). "Lag" generally matches with "colluvium: (Related Match). However, when described in the comments, it can be associated with its protolith which results into a Failed Match as "rock".




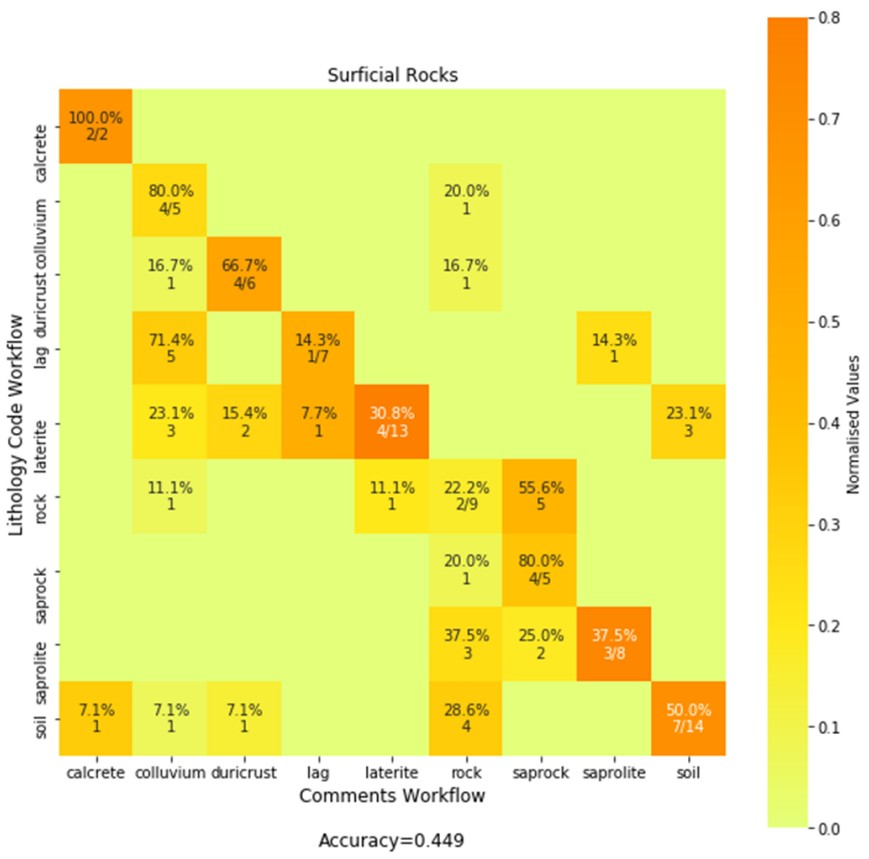

| | precision | recall | f1-score | support |
|---|---|---|---|---|
| calcrete | 0.67 | 1.00 | 0.80 | 2 |
| colluvium | 0.27 | 0.80 | 0.40 | 5 |
| duricrust | 0.57 | 0.67 | 0.62 | 6 |
| lag | 0.50 | 0.14 | 0.22 | 7 |
| laterite | 0.80 | 0.31 | 0.44 | 13 |
| rock | 0.17 | 0.22 | 0.19 | 9 |
| saprock | 0.36 | 0.80 | 0.50 | 5 |
| saprolite | 0.75 | 0.38 | 0.50 | 8 |
| soil | 0.70 | 0.50 | 0.58 | 14 |
| macro avg | 0.53 | 0.53 | 0.47 | 69 |
| weighted avg | 0.57 | 0.45 | 0.45 | 69 |

**Figure 15. Confusion matrix for surficial comparing the fuzzy string matching results from the Lithology_Code workflow (vertical axis) and Comments workflow (horizontal axis). The heatmap shows the values normalised to the support size to address the imbalance between classes. The values shown in the cells indicate the number of samples classified for the class. Empty cells indicate zero samples. The accuracy is 44.9%, with a weighted average precision of 57% and recall of 45%. These results were taken from a subset of 69 samples across 9 classes.**





## 6    Discussion

**6.1 Data Extraction**

*dh2loop* supports data extraction of collar, survey and lithology interval tables. The main consideration in the data extraction was that the data retrieved was complete, relevant and useful. We would rather throw erroneous or questionable data out and have the rest with a high level of confidence, than the other way around. 93% of the available collar data in the area was extracted successfully. This can be improved by implementing alternative ways for retrieving `RL` and `MaxDepth` values. For

example, if no `RL` values are fetched from the database, it could be fetched from open-source digital terrain models (DTM) and/or SRTM (Shuttle Radar Topography Mission). As for missing `MaxDepth` values, the maximum `ToDepth` values in the survey and/or interval tables could be used.

The survey extraction rate of 86% was fairly good. *dh2loop* ensures that the `Azimuth` and `Dip` values are sensible

measurements before including them into the extracted output file. An improvement that could be implemented is to run an assessment on the deflection angles for each drill hole and flag intervals with unrealistic deflection angles.

The lithology extraction using the Lithology Code workflow shows that the bottle neck to its extraction rate is the extensiveness of the Lithology Code thesaurus. Since the thesaurus did not have the information for all companies in the area, only 34% of

the available information was retrieved. The extraction results for the Comments workflow cannot be compared with the Lithology Code workflow as only the intersection of both workflows was considered in this study.

**6.2 Thesauri**

*dh2loop* provides the user with 9 thesauri that deal with the extraction of collar, survey and lithology interval tables. For extraction of other properties, such as downhole alteration, geochemistry, mineralogy and structures, at least one thesaurus is

needed for each attribute we would like to export. These thesauri are built manually by inspecting all the terminologies available in the database. Although, creating them can be tedious, updating an existing thesaurus is as simple as adding and/or removing a word to the list. There are many other properties available in the database that could be exploited using the existing methodology, thus there is an incentive in finding a way to improve the methodology of building these thesauri. Analysis on the syntax of the existing thesauri may help in automating creation of other thesauri.


The Hierarchical Lithology thesaurus puts equal weight on each of the entries in the thesaurus. Knowing the geology in a user's area, the matching can be improved by adding more weight to prevalent lithologies through adding a bonus score.

**6.3 Assessment of String Matching Results**

The string matching results highlights that geological drill core logging is prone to human error and bias, and result to incorrect

logs. Sometimes even if the data is available and correct, it is not in format that can be directly extracted. For example, the `Comment` field/s are filled with a string description such as "same as above" and "-do-". Currently, for this case, *dh2loop* returns without a match. In the future, we could be able to search through the previous entries to retrieve the correct lithology. Furthermore, the code does not handle and check for inconsistencies in the logs. It only addresses the inconsistencies in nomenclature and not the logging itself. The string matching misclassification results illustrate that importance in the

consistency and level of detail being put into logging and identifies differences in convention or uncoordinated logging among geologists. *dh2loop* provides a notebook that demonstrates using *striplog* to improve the consistency of the logs through data pruning and annealing. In the future, the geochemical compositions can be used to counter check and lithology assigned to the interval.





Comparing the string matching between the Lithology Code and Comments workflow, the Lithology code workflow results to a higher matching rate, 86% of the extracted data is successfully matched. Comparing this subset to the Comments workflow, the matching rate is much lower at 16%. This shows that the Lithology Code workflow, while potentially tedious, results into a higher percentage of successful matches. However, if we are considering a regional study involving multiple companies and drilling campaigns, building thesauri can be time-consuming depending on the size of the region being studied,

number of attributes of interest, number of companies and drilling campaigns. This could range from a couple of hours to months. It can also be tedious as it involves inputting errors and inconsistencies as well as exhausting all permutations for decision-tree based logging systems

Comment matching provides a quicker way to standardize and classify rocks. The comprehensive clean-up dictionary allows

assists in improving the matching accuracy. Given the context that we are dealing with legacy data, an extraction rate of 16% from the Comments is not bad at all. With minimal effort, we obtain additional geological data wherein, although of a smaller percentage (31% of Exact Matches) but with reasonably high confidence in its quality. It is important to note that most of the time Failed Matches are not a result of the limitations of the algorithm but of the information being fed itself. Inconsistent logs (`Company_Litho` data is different from `Comment`) usually occur when:

1. The logs were post-processed and correlated with the rest of the hole or neighbouring drill holes and changes were made to the `Company_Litho` but none on the `Comments` field.

    2. The `Comments` would have more level of detail than the `Company_Litho`. In this case, we may get a lithology at `Lithology_Subgroup` from the Lithology Code workflow and a `Detailed_Lithology` from the Comments workflow.

3. The `Company_Litho` would have more level of detail than the `Comments`

    4. `Comments` contains the description of the whole intercept, which could include a contact of two lithologies or intercalating lithologies.

From the results of the confusion matrix (Sect. 5), some rock groups are more sensitive to these inconsistencies than others.

There is higher confidence in the classification of structures and textures and metamorphic rocks. The user should be more careful when dealing with igneous, sedimentary and surficial rocks. They are more difficult to classify as the way they are described are highly variable between different geologists. For structure-related lithological descriptions the small number of misclassifications occur where faults, veins and fillings coexist. For metamorphic rocks, entries like "mica amphibolite schist" can cause Broader Matches with the confusion of whether to classify it as "amphibolite" or "schist". "Schist" is a textural term

of medium grade metamorphic rock with a medium to coarse-grained foliation defined by micas while "amphibolite" is a compositional term representing a granular metamorphic rock which mainly consists of hornblende and plagioclase. One should be wary about these possibilities as they may impact the interpretation of the geology in the area. For sedimentary rocks, the lack of a standard syntax as to how comments are recorded impacts the classification. Descriptions of intercalated lithologies or presence of major and minor lithology can result to Failed Matches. Igneous rocks perform fairly well, most of

what is not captured as Exact Matches are captured at least as Broader Matches. These are usually related to either an inconsistent level of detail between the fields or rock types used as descriptors ("komatiitic", "andesitic", basaltic").

Low matching accuracy in surficial rocks can be attributed to the lack of universally agreed terminology for: deeply weathered regolith; poorly-defined and misapplied surficial rock nomenclature; wide range and variation of materials within the regolith

and; difficulty in bulk mineral identification from macroscopic samples. Furthermore, since the degree of weathering of minerals generally increases from the bottom to the top of in-situ weathering profiles, the intermixing of strongly weathered



and less weathered grains may cause confusion (Cockbain, 2002). Ubiquitous, highly variable and less interesting lithologies also cause mismatches. An example of this is "soil". Soils are technically are not rocks but is commonly used in logs to refer to the first intercept of the regolith or to describe highly weathered, clay-rich and unidentifiable intercept. Soils vary in

character from thin, coarse-grained, poorly differentiated lithosols to thick, well-differentiated silt and clay-rich soils. Soils were classified with the highest variability of terms: "soil", "rock", "duricrust", "colluvium" and "calcrete". There are also certain lithologies with ambiguous nomenclature conventions, like "laterite", "duricrust", "lag". Some geologists use laterite to refer to the whole lateritic profile (ferruginous zone, mottled zone, and saprolite) while others to refer to the ferruginous zone (Eggleton, 2001). Ironcrust, duricrust, lateritic gravels and lag are commonly used interchangeably. Duricrust and

ironcrust are terms to describe ferruginous indurated accumulations at or just below the surface. The difference in usage of the term laterite and the interchangeability of duricrust and lag explains the misclassification of "laterite" to "colluvium", "duricrust" and "lag". Another example is "saprolite" and "saprock". They are ambiguous terminologies as they both represent the lower horizons of lateritic weathering profiles, with saprolites having more than 20% of weatherable minerals altered and saprock having less than 20% of the weatherable minerals being altered (Eggleton, 2001). This arbitrary limit makes the

terminology used in the logs easily interchangeable, thus affecting the `Detailed_Lithology` matching.

Ideally, a combination of the Lithology Code and Comments workflow should result in a more robust classification. This will also allow the user to have a better look at the result of both workflows and decide what is appropriate for one's purpose.

### 6.4 Value of the Lithological Information Extracted for Multiscale Analyses

The *dh2loop* lithology export provides a standardized lithological log across different drilling campaigns. This information can readily imported into 3D visualization and modelling software. This allows for drill hole data to be incorporated into 3D modelling, providing better subsurface constraints, especially at a regional scale. It also allows the user to decide on the lithological resolution necessary for their purpose. It provides a three-level hierarchical scheme: `Detailed_Lithology`, `Lithology_Subgroup` and `Lithology_Group`  that can be used as an input to multiscale geological modelling.

*dh2loop* can be improved by correlating the these lithologies to their corresponding stratigraphic formations. Having the spatial extents of the different geological formations and their lithological assemblages (GSWA Explanatory Notes System) as well as a couple of stratigraphic drill holes, it may be possible to infer the corresponding stratigraphic formation.

### 6.7 *dh2loop* Functions and Notebooks

The *dh2loop* library supports a workflow that extracts, processes and classifies lithological logs (Appendix A3). This library

was built to extract drill hole logs from the WAMEX database. The assumptions made in the entire workflow attempts to replicate the thought process of a geologist performing the data extraction, data quality checks and lithological log classification manually. However, it can be adapted for other geological relational databases or from other table formats. An example using comma separated values tables (CSVs) is shown in the notebook: Exporting and Text Parsing of drill hole Data Demo.


In addition to the data extraction, downhole desurveying and lithological matching functions discussed, *dh2loop* also provides functionalities and a notebook demonstrating harmonization of drill hole data. This is useful for combining and correlating drill hole exports of different properties such as lithology, assays and alteration. It is also possible to export this information in Visualization Toolkit format (.VTK). It also provides a notebook that demonstrates the application of *lasio* and *striplog* on

*dh2loop* interval table exports. WAMEX reports can also be interactively downloaded through a notebook provided in the package.



## 7 Conclusions

The *dh2loop* library is an open-source library that extracts geological information from a legacy drill hole database. This workflow has the following advantages:

1. Maximizes the decades of legacy geoscientific data available

2. Gains better subsurface characterization, where data is available

3. Standardizes geological logs across different drilling campaigns, a necessary but typically time-consuming and error-prone activity

4. Provides s set of complementary thesauri that can easily be updated

5. Provides additional subsurface constraints which are critical for 3D geological modelling

6. Implements a hierarchical classification scheme that can be used as an input to multiscale geological modelling

7. Classification results can also be used as a tool to improve future geological logging works by revealing common errors and sources of inconsistencies



**Code and Data Availability**

*dh2loop* is a free, open-source python library licensed under the MIT License. It is hosted on the GitHub repository https://github.com/Loop3D/dh2loop and can be cited as http://doi.org/10.5281/zenodo.4043568.

**Author Contribution**

M. Jessell contributed the original idea, which was further developed by R. Joshi. K. Madaiah developed the code. M. Jessell, M. Lindsay, G. Pirot provided guidance and direction in the research. R. Joshi prepared the manuscript with contributions from all co-authors. Lastly, M. Jessell supervised the entire process.

**Acknowledgements**

The research was carried out while the first author was supported in receipt of Scholarship for International Research Fees (Australian Government Research Training Program Scholarship) and Automated 3D Geology Modelling PhD Scholarship (University Postgraduate Award) at the University of Western Australia. The work has been supported by the Mineral Exploration Cooperative Research Centre whose activities are funded by the Australian Government's Cooperative Research Centre Program. This is MinEx CRC Document 2020/***. This work was also done with the Loop Consortium (http://loop3d.org) as part of an international effort to found a new open-source platform to build the next generation of 3D geological modelling tools. Mark Lindsay is funded by ARC Discovery DE190100431. We would also like to acknowledge Tim Ivanic for his inputs on the geology of the Yalgoo-Singleton greenstone belt.





**Appendix A:** *dh2loop* **package information**

**A1 Installation**

Installing *dh2loop* can be done by cloning the GitHub repository with $ git clone https://github.com/Loop3D/dh2loop.git and

then manually installing it by running the python setup script in the repository: $ python setup.py install

**A2 Documentation**

*dh2loop's* documentation provides a general overview over the library and multiple in-depth tutorials. The tutorials are provided as Jupyter Notebooks, which will provide the convenient combination of documentation and executable script blocks in one document. The notebooks are part of the repository and located in the notebooks folder. See http://jupyter.org/ for more

information on installing and running Jupyter Notebooks.

**A3 Jupyter notebooks**

Jupyter notebooks are provided as part of the online documentation. These notebooks can be executed in a local python environment (if the required dependencies are correctly installed). In addition, static versions of the notebooks can currently be inspected directly on the *github* repository web page or through the use of *nbviewer*.

1.  WAMEX Interactive report downloads
        (https://github.com/Loop3D/dh2loop/blob/master/notebooks/0_WAMEX_Downloads_Interactive.ipynb)

    2.  Exporting and text parsing of drill hole data from PostgreSQL database
        (https://github.com/Loop3D/dh2loop/blob/master/notebooks/1_Exporting_and_Text_Parsing_of_Drillhole_Data_Fr
om_PostgreSQL.ipynb)

3.  Exporting and Text Parsing of drill hole Data Demo
        (https://github.com/Loop3D/dh2loop/blob/master/notebooks/2_Exporting_and_Text_Parsing_of_Drillhole_Data_D
emo.ipynb)

    4.  Harmonizing drill hole data
        (https://github.com/Loop3D/dh2loop/blob/master/notebooks/3_Harmonizing_Drillhole_Data.ipynb)


**Appendix B: Thesauri**

A few examples of each thesauri are shown below. Each thesauri includes alternate nomenclature and spelling. The complete thesauri are available at: https://github.com/Loop3D/dh2loop/blob/master/thesauri/

**B1 Drill hole Collar Elevation (https://github.com/Loop3D/dh2loop/blob/master/tThesauri/thesaurus_collar_elevation.csv )**

1. """RL"""
2. ADJ_RL
3. Adjusted_RL
4. AMG_mRL
5. Approx RL
6. Arbitary_RL
7. Best_RL
8. COLL_RL
9. Collar Elevation
10. Collar RL
11. Collar RL (m)
12. Corrected_RL
13. DB_RL
14. DEM_RL
15. DGPS Elevation MGA94 Zone 51
16. DGPS RL
17. DTM RL
18. Elevaion
19. Elevat
20. Elevati
21. Elevatio
22. Elevation
23. elvation
24. Lidar_RL
25. Local RL
26. MGA RL
27. MGA_Elev
28. MGA_Elevation
29. MGA_RL_Z
30. MGA_Z50_RL
31. MGA94_RL
32. MINE_RL
33. mRL
34. NAT RL
35. Orig RL
36. 
37. Orig_Reg_RL
38. R.L
39. R.L.
40. Raw_RL
41. Real RL
42. ref_mRL
43. RL
44. Surveyed RL
45. UTMElev
46. UTMmRL
47. WGS84_WORLD_LL_Calc_Z
48. z RL
49. Z_(RL)
50. ZCOLLAR_RL
51. ZMINE_RL

**B2 Drill hole Maximum Depth Thesaurus (https://github.com/Loop3D/dh2loop/blob/master/thesauri/thesaurus_collar_maxdepth.csv)**

1. """DEPTH"""
2. "Depth
3. """Final
4. AC Depth
5. Actual Depth
6. DD_Depth
7. DDH_Depth
8. Depth
9. Depth m
10. DEPTH (m)
11. Depth (EOH)
12. Depth (metres)
13. Depth_D
14. Depth_DD
15. Drill Depth
16. Drilled_Depth
17. End of hole depth
18. END_DEPTH
19. EOH Depth
20. EOH Depth (Metres)
21. F/Depth
22. F/Depth(m)
23. F_Depth
24. FIN_DEPTH
25. Final Depth
26. Final Depth (m)
27. Final_dpth
28. Finl_Depth
29. HDEPTH
30. Hole depth
31. MAX DEPTH
32. max_depth (m)
33. Max_Depth m
34. Maximum Depth
35. T_depth
36. TD
37. toatl_depth
38. TOT DEPTH
39. TOT_DEPTH_M
40. Total Depth
41. Total Depth (m)
42. Total Depth Drilled m
43. Total Depth M
44. Total Hole Depth



**B3 Drill hole Survey Azimuth Thesaurus**
(**https://github.com/Loop3D/dh2loop/blob/master/thesauri/thesaurus_survey_azimuth.csv** )

| | | | | | |
|---|---|---|---|---|---|
| 1. | AMG AZIMUTH | 1085 | 22. | Azimuth Local | 43. Magnetic  AZI |
| 2. | AMG_azim | | 23. | Azimuth Mag | 44. MGA AZI |
| 3. | AMGAZM | | 24. | Azimuth(T) | 45. MGA Azimuth |
| 4. | Aximuth | | 25. | AzLoc | 46. MGA94_Az |
| 5. | Aximuth_gyro | | 26. | GDA_Az | 1110 | 47. NAT_Azimuth |
| 6. | AZ | 1090 | 27. | GILBEYS AZI | 48. Nominal Az |
| 7. | Az_AMG | | 28. | GRID AZI | 49. Nominal AZI |
| 8. | Az_grid | | 29. | GRID_AZ | 50. Orig AZI |
| 9. | AZ_LOCAL | | 30. | GridAzim | 51. Orig Azimuth |
| 10. | AZ_Mag | | 31. | LOCAL AZIMUTH | 1115 | 52. Orig_Azim |
| 11. | AZ_MINE | 1095 | 32. | Local_Az | 53. ORIG_AZIMU |
| 12. | AZI GRID | | 33. | Local_Azi | 54. Project_Azim |
| 13. | AZI(T) | | 34. | Local_Azim | 55. Ref. AZI |
| 14. | Azi_Mag | | 35. | LOCALAZID | 56. ref_azim |
| 15. | AZI_MGA | | 36. | LocAzim | 1120 | 57. REG_AZIM |
| 16. | AZIM | 1100 | 37. | LOCAZM | 58. UTM_Az |
| 17. | Azim mag | | 38. | Mag_Az | 59. UTM_Azi |
| 18. | Azim_AMG | | 39. | Mag_Azim | 60. UTM_Azimuth |
| 19. | Azim_Local | | 40. | Mag_Azimu | 61. WMC AZI |
| 20. | Azim_M | | 41. | MagAzi | |
| 21. | AZIMUTH | 1105 | 42. | MAGAZM | |

**B4 Drill hole Survey Dip Thesaurus**
(**https://github.com/Loop3D/dh2loop/blob/master/thesauri/thesaurus_survey_dip.csv** )

1. Dip
2. Dip (deg.)
3. Dip_2
4. INC
5. Inclination
6. DIP_camera
7. Dip_gyro
8. DIP_LOCAL
9. Nominal_Dip
10. DIP_Surtron

**B5 Drill hole Lithology Thesaurus**
(**https://github.com/Loop3D/dh2loop/blob/master/thesauri/thesaurus_geology_lithology.csv** )

| | | | | | |
|---|---|---|---|---|---|
| 1. | $Lith | | 5. | 1_LithCode | 9. Geol |
| 2. | %Maj | 1145 | 6. | 1_RootCode | 10. Geol Code |
| 3. | %Maj_Lith | | 7. | F_lithology | 1150 | Geol General |
| 4. | %Major Lith | | 8. | GeoCode | 11. GEOL_Rock1 |





12. geol_type
13. GEOL1
14. Geological Unit
15. Geological_code
16. GEOLOGY
17. Geology Code
18. HOST_LITH1
19. Intermediate Rock Type
20. lihological
21. Lilh1
22. lit
23. Lit_1
24. lith
25. Lith   Code
26. Lith 1
27. Lith 1 Rock Type
28. Lith Maj
29. Lith Major
30. Lith_Cat
31. Lith_Codes
32. Lith_Maj_1
33. Lith_PrimaryCode
34. LITH_Protolith
35. LITH_TYPE
36. Lith01
37. lith-1

38. Lith1_Code1
39. LithCode1_A
40. lithcode1_main
41. LithCode1_T
42. LithCode1_V
43. LithCodeSy
44. Lithgen
45. LithGenrl
46. Lithic_code
47. Lithlogy 1
48. LithMajor1
49. LithMin1
50. Litho
51. LITHO _1
52. Litho Code
53. Litho Type
54. LITHO_PLOT
55. Lithological Unit
56. LITHOLOGY
57. Lithology_rock
58. Lithoology
59. Main Geol Unit
60. Main Lithology
61. MAJ LITH
62. Maj Lithcode
63. Maj. Lithology

64. Maj. Rock
65. Major Lithology
66. Major Rock
67. Major Rock Type
68. Primary Lith
69. Primary RockType
70. ROCK
71. ROCK CODE
72. Rock Group
73. ROCK NAME
74. Rock type code
75. Rock Type Major
76. Rock Unit
77. Rock_id
78. ROCK-CODE
79. RockLithCode
80. RockMain
81. RockMajor
82. Root_Code
83. Root_Lith
84. Root_rock
85. Wiluna Lithology Code
86. WMC ROCK Code
87. wmc_lith1

**B6 Drill hole Comments Thesaurus**
(**https://github.com/Loop3D/dh2loop/blob/master/thesauri/thesaurus_geology_comment.csv** )

1. Comment
2. COMMENTS
3. D_stLITHCOMMETNS
4. Description
5. INTRCPT_COMMENT
6. LITH_COMMENT

**B7 Drill hole Lithology Codes Thesaurus**
(**https://github.com/Loop3D/dh2loop/blob/master/Thesauri/thesaurus_geology_lithology_code.csv** )

**CompanyID > Company_LithoCode > Company_Litho**

1. 551 > BIF > Banded iron formation
2. 1311 > BIF > Banded iron formation
3. 551 > CS > Saprolite, undifferntiates
4. 2551 > Cs > Cambrian Sediment

5. 551 > CSM > Mafic saprolite
6. 2551 > Csm > Cambrian Sediment Limestone
7. 2551 > Cv > Cambrian Vein material
8. 2790 > CV > Colluvium



9. 551 > CY > Clay, undifferentiated

10. 2551 > Cy > Cambrian Mylonite

11. 2551 > Hp > Carboniferous Massive sulphide

12. 2790 > HP > Hardpan

13. 2551 > Lc > Oligocene Chemical Sediments

14. 3053 > Lc > undiferentiated laterite clay

15. 2551 > Ls > Oligocene Sediment

16. 3053 > Ls > undiferentiated laterite sand

17. 1311 > MB > Basalt

18. 2551 > Mb > Miocene Mafic Extrusive

19. 369 > PH > Phyllite

20. 2551 > Ph > Proterozoic Hornfels

21. 2551 > Qc > Quaternary Chemical Sediments

22. 3053 > Qc > undifferentiated recent surficial deposit clay

23. 2551 > Qs > Quaternary Sediment

24. 3053 > Qs > undifferentiated recent surficial

deposit sand

25. 551 > QV > Quartz Vein

26. 2551 > Qv > Quaternary Vein material

27. 369 > S > Undifferentiated sediments

28. 3053 > S > undifferentiated sediment

29. 2551 > Sh > Silurian Hornfels

30. 3053 > Sh > shale

31. 551 > SS > Sandstone

32. 2551 > Ss > Silurian Sediment

33. 369 > TCC > Channel clays

34. 1311 > TCC > Tertiary palaeochannel clay

35. 369 > TCS > Channel sands

36. 1311 > TCS > Tertiary palaeochannel clayey sand

37. 369 > TLC > Lake clays

38. 2551 > Tlc > Triassic Lamprophyre/Kimberlites

Carbonatite

39. 369 > ALL > Alluvium

40. 2790 > ALLU > Alluvium

41. 2790 > AMPH > Amphibolite

42. 369 > MAA > Amphibolite

43. 369 > MBA > Basalt

44. 2790 > CALC > Calcrete

45. 369 > LCZ > Calcrete

46. 11410 > CL > Clay

47. 2790 > CLAY > Clay

48. 369 > COL > COLLUVIUM

49. 2790 > COLL > Colluvium

50. 369 > FDI > Diorite

51. 2790 > IDO > Diorite

52. 3053 > Bdo > dolerite

53. 2790 > DOLR > Dolerite

54. 1311 > MD > Dolerite

55. 3053 > Bgb > gabbro

56. 2790 > GABR > Gabbro

57. 2790 > MDG > Gabbro

58. 11410 > MG > Gabbro

59. 369 > MGA > gabbro

60. 369 > FGR > Granite

61. 11410 > GR > Granite

62. 2790 > GRAN > Granite

63. 2790 > HARD > Hardpan

64. 369 > LDH > Hardpan

65. 2790 > NMO > Mottled zone

66. 2790 > WMZ > Mottled zone

67. 369 > FPG > Pegmatite

68. 2790 > PEGM > Pegmatite

69. 2790 > QZVN > Quartz Vein

70. 2790 > VQZ > Quartz Vein

71. 2790 > FSS > Sericite schist

72. 551 > SESCH > Sericite Schist

73. 2790 > UMS > Serpentinite

74. 3053 > Us > serpentinite

75. 369 > SSH > Shale

76. 369 > LSZ > Silcrete

77. 2790 > SILC > Silcrete

78. 2790 > SLS > Siltstone

79. 369 > SSTS > Siltstone

80. 369 > M > Undifferentiated Mafic Rocks

81. 2790 > MOO > Undifferentiated mafic rock



**B8 Clean up dictionary (https://github.com/Loop3D/dh2loop/blob/master/thesauri/thesaurus_cleanup.csv)**

**B9 Lithology Hierarchical Thesaurus
(https://github.com/Loop3D/dh2loop/blob/master/thesauri/thesaurus_geology_hierarchical.csv )**



**Appendix C:**

**C1 Configuration file**

```
#Extents to query
minlong=115.5
maxlong=118
minlat=-30.5
maxlat=-27.5

#src_pro,Dst_proj
src_csr = 4326
dst_csr = 28350
#ExportFiles
export_path='../data/export_db/'
DB_Collar_Rl_Log = export_path + 'DB_Collar_Rl_Log.log'
DB_Collar_Maxdepth_Log = export_path + 'DB_Collar_maxdepth_Log.log'
DB_Collar_Export=export_path+'DB_Collar_Export.csv'
DB_Survey_Export=export_path+'DB_Survey_Export.csv'
DB_Survey_Export_Calc=export_path+'DB_Survey_Export_Calc.csv'
CET_Litho=export_path+'CET_Litho.csv'
DB_Lithology_Export=export_path+'DB_Lithology_Export.csv'
DB_Lithology_Export_Backup=export_path+'DB_Lithology_Export_Backup.csv'
DB_Lithology_Upscaled_Export=export_path+'DB_Lithology_Upscaled_Export.csv'
Upscaled_Litho_NoDuplicates_Export = export_path+'Upscaled_Litho_NoDuplicates_Export.csv'
DB_Lithology_Export_Calc=export_path+'DB_Lithology_Export_Calc.csv'
DB_Lithology_Export_VTK=export_path+'DB_Lithology_Export.vtp'
print('Default parameters loaded from DH2_LCConfig.py:')
with open('../notebooks/DH2_LCConfig.py', 'r') as myfile:
  data = myfile.read()
  print(data)
myfile.close()
print('\nModify these parameters in the cell below')
```



**C2 Fuzzy String Matching Pseudocode**

```
DEFINE FUNCTION Attr_val_With_fuzzy():
       Bestmatch is -1
       Bestlitho is ' '
       list top
       I is 0
list    +attr_val_sub_list

       open csv file Attr_val_fuzzy for writing .
       write csv file heading CollarID','code','Attr_val','cleaned_text','Fuzzy_wuzzy','Score'

Convert list Var.Attr_val_Dic to  list of list Attr_val_Dic_new
       For each element Attr_val_Dic_ele in Attr_val_Dic_new
               cleaned_text_1= call clean_text user defined function with attribute_value
               cleaned_text_1=call tokenize_and_lemma python function with cleaned text
               cleaned_text_1
cleaned_text=join each word of cleaned_text_1 with space as one string.
               Words is replace slashes by space using re pattern, strip leading and trailing spaces.
               Words  is strip \n\r and split on space.
               for each element Litho_dico_ele in Var.Litho_dico
                       litho_words is Litho_dico_ele with lower case, rstrip \n\r,replace ( or) by space and split
on space
                       scores  =  call  python  process.extract  with  arguments  cleaned_text,  litho_words,
                       scorer=fuzz.token_set_ratio
                       for sc in each scores
                               if(sc[1]>bestmatch): #better than previous best match
bestmatch =  sc[1]
                                       bestlitho=litho_words[0]
                                       top.append([sc[0],sc[1]])
                                       if(sc[0]==words[last]): #bonus for being last word in phrase
                                               bestmatch=bestmatch*1.01
elif (sc[1]==bestmatch): #equal to previous best match
                                       if(sc[0]==words[last]): #bonus for being last word in phrase
                                               bestlitho=litho_words[0]
                                               bestmatch=bestmatch*1.01
                                       else:
top.append([sc[0],sc[1]])
               if bestmatch >80:
                       write bestlitho and bestmatch along with required data to csv file
                       clear top
                       CET_Litho as ' '
Bestmatch  as -1
                       Bestlitho as ''
               else
                       write 'Other' as bestlitho with bestmatch along with required data to csv file
                       clear top
CET_Litho as ' '
                       Bestmatch  as -1
                       Bestlitho as ''
```

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
