# Peer review of "dh2loop 1.0: an open-source Python library for automated processing and classification of geological logs"

_Geoscientific Model Development, 2020_

## Author Comment (AC1)

**Response to Anonymous Referee #1:**

The breakdown of replies to the individual comments are below:

**General comments**

The paper is too long and dense in terminology and nuanced meaning. The authors have endeavoured to symbolise some of the different and complicated database terminology although possibly more is needed. The flow is logical and the writing is generally understandable although laboured in places. The figures and tables are appropriate and informative in most places; some minor improvements have been suggested. I have not checked references or URL links exhaustively.
→ In the updated version, we reduced the length of the paper by putting the details of Section 2.4.2 Survey Extraction in the Appendix. We addressed the dense terminology by defining all the variables in a table in Section 2.1. Conventions. We symbolized the workflows differently as you have suggested.

**Specific comments**

Introduction, section 1: replace first 2 sentences with 'Drilling is a process of penetrating through the ground that is capable of extracting information about rocks from various depths below the surface. This is useful for establishing the geology beneath. Drill core or cuttings can be collected thus providing samples for description, interpretation and analysis.'
→ This was revised in the updated version.

Introduction, section 3: The legacy data described seem to be hardcopy forms subsequently digitised. Legacy digital data also suffer from lack of standardisation, inconsistency.
→ Legacy digital data is included and this correction is revised in the updated version.

Introduction, line 78: These data are not 'unstructured' but they may not conform to standards or be consistently applied/described.
→ Unstructured data has an imprecise definition across different references. We agree there is some ambiguity as the data does reside in a relational database (as structured data is). However, the data we are dealing with requires text analysis to sort and extract data. In this study, we refer to "unstructured" to mean that the written content cannot be readily mapped onto standard database fields and not easily searchable. In the case of the free text comments, descriptors such as color, age, texture are not written in any order or using any standard. In the manuscript, we added description as to what we mean by "unstructured" for clarity, while acknowledging the potential ambiguity.

Material and Methods, Conventions: workflows need their own distinct convention (font). Later they are confusingly rendered as combinations of database table fields.
→ This was revised in the updated version.

Thesauri: Some of the so-called 'synonyms' actually have distinct meanings from each other, even the listed example elevation vs relative (reduced) level. Maybe qualify 'synonym' as meaning 'nearly the same' or a 'close match' for their general intent is similar e.g the elevation terms all are recording a vertical height.
→ Definition of "synonyms" in this work is expounded in the updated version.

Thesauri, line 245: Rather than 'The opposite is true as well' suggest you explain specifically that more than one code may refer to the same lithology. Basically there is a many-to-many relationship between code and lithology.
→ We agree and applied this in the updated version.

Thesauri, line 252: The CGI vocabularies support the GeoSciML and EarthResourceML (note singular) geology data models but potentially other applications. Suggest you rephrase as '...the CGI-IUGS geoscience vocabularies accessible at http://geosciml.org/resource/def/voc/'.
→ This was revised in the updated version.

Thesauri, Lithology Hierarchical Thesaurus: The 3-level hierarchy is highly simplified compared to CGI's Simple Lithology. Many of the 'Lithology_Subgroups' listed have parent-child relationships e.g. 'mafic_fine_grained_crystalline' is a child of 'mafic'. This should be mentioned, presumably some simplification and pragmatism is needed for your analysis.
→ Parents in parent-child relationships are included with their children as catch-all groups to capture free text descriptions that do not include details that would be captured by only using the child term alone.

Data Extraction, Collar Extraction: I wondered why this section needs to be here at all. The collar extraction isn't central to your paper focus on lithology. You don't utilise collar location in a spatial analysis or context - its only function in this paper seems to be a pre-filter for data quality. The method itself is good and useful, and ultimately important for data mining where spatial understanding is needed, just not essential for the lithology-driven analysis presented here.
→ We understand the concern with having this section in the paper. However, while the drill collar may appear to be a trivial example, it is to prime the reader for the more complicated following sections. We were using the collar extraction as an introduction to the database structure before transitioning to the database structure for lithology which also has two workflows. This is also to emphasize that dh2loop provides the three basic interval tables needed to import geological data into 3D modelling softwares.

Data Extraction, Survey Extraction: Ditto, I wondered why this section needs to be here at all. If it is retained then suggest the 4th field should be 'Inclination' not 'Dip'. Dip is a measurement of the slope of a planar surface feature whereas inclination refers to the plunge of a linear feature. Additionally, how consistent are WAMEX records around positive inclinations meaning upwards-directed drill holes?
→ We addressed this by putting the bulk of the Survey extraction as part of the Appendix. The term "dip" is also changed into "inclination" in both the paper and the code. In this dataset, we obtained 1014 dip values that are between 0 to 90 (from 684 holes, 23 companies, 28 reports). These examples of underground holes drilled upwards are accommodated in the workflow, so long as the metadata and data appropriately describe them as such. Detecting where an upwards directed hole is when not reported as such is beyond the scope of the paper, and second-guesses the database.

Data Extraction, Survey Extraction: The 'Calculated X, Y, Z values' are not particularly helpful or necessary in this Survey table i.e. only recording collar and the end of hole locations. Survey tables more typically describe changing azimuths and inclinations with 'depth' (i.e. account for curved drill holes) – does WAMEX not do this?
→ WAMEX does describe changing azimuths and inclinations with depth. The Calculated X, Y, Z is a functionality made available as it is also used in the Lithology Table. This is shown here as some software accept calculated XYZ as input parameters for survey. Since we moved this section to the Appendix, we think it be alright to include this detail.

Data Extraction, Lithology Extraction: The fuzzywuzzy algorithm appears to be repeating pre-processing already mentioned in the previous paragraph (line 419-424).
→ The pre-processing cleans up the text, while the fuzzywuzzy algorithm matches the text to a dictionary. We added this text to the introduction of Section 2.4.

Data Extraction, Line 432-433: What does 'Since the sorted intersection component of token_set(), will result in an exact match...' mean? Elaborate or explain more clearly.
→ Sorted intersection tokens are the similar tokens (characters) between the two strings, thus it always equates to an exact match (=100). The remainder component is what lowers the score. We included this in the text.

Data Extraction, Line 439: What is an 'intersection token'?
→ Intersection token are the similar tokens between the two strings. We added this into the text.

Data Extraction, Table 1: The column of ticks and crosses is unexplained. In two cases the lower score is ticked implying it is the preferred result?
→ Yes, we expanded the caption to clear this. This was revised in the updated version.

Data Extraction, Line 453: 'Andesitic basalt' is an unfortunate example since 'basaltic andesite' is an established volcanic rock name. Would basaltic andesite wrongly revert to andesite in this classification process?
→ This raises a good point. We suggest to sticking with the "andesitic basalt" example, but we further explained that in the case of "basaltic_andesite" it will not be simplified into andesite, as the thesaurus includes an entry for basaltic_andesite, since it is an established volcanic rock name as you have mentioned.

Data Extraction, Figure 8: I struggled to understand this graph. How can data with 100% Exact Match score only 80%?
→ The graph shows that using a smaller dataset, we could notice that using a score cut-off of 80, ~99% are returned as exact matches. This exercise tests at what cut-off score does the number of exact matches plateau. This is to avoid using a very stringent 100 cut-off to capture Exact Matches. This cut-off score is a parameter that can be changed and is dependent on the data being processed. We added this into the figure caption for more clarity.

Data Extraction Results, Unique Lithology Code Results: Database table field names seem to be inconsistent e.g. Company_LithoCode vs Company_Lithology vs Lithology_Code. Suggest careful check to ensure consistency of use otherwise confusing for the reader.
→ This was revised in the updated version.

Data Extraction Results, Unique Lithology Code Results: Workflows such as 'Lithology_Code Detailed_Lithology' need to be distinctly symbolised. At the moment they look like co-joined database table names without an obvious algorithm progression between them. Suggest also where these are mentioned you mention 'workflow' or 'workflows'
→ This was revised in the updated version.

Data Extraction Results, Table 2: Struggling to understand why row 3 is a Close Match when the almost identical row 5 is a Broad Match? If anything the 'basic volcanic rock', not being a recognised Lithology_SubGroup member, is broader rather than closer than 'mafic fine grained crystalline'.
→ We agree with the comment. We changed "basic volcanic rock" to "basaltoid" to better illustrate the difference.

Data Extraction Results, Fuzzy String Matching Results: Mentions of 'comments' should be 'Comments' in most cases, possibly with special font.
→ This was be revised in the updated version.

Data Extraction Results, lines 611, 622: These results are suboptimal. You discuss this later but it seems your method is sometimes picking a subordinate lithology rather than the dominant lithology.
→ We kept the discussion of the results as part of the discussion.

Data Extraction Results, line 653: I wasn't clear what 'the limitation' is – processing?
→ The limitation we are describing is that it is not possible to compare the matches for the cases where only one of the workflows arrive with a match. We expounded on this in the updated version manuscript.

Discussion, Assessment of String Matching Results (line 844): Need to qualify that the 'classification of structures and textures and metamorphic rocks' has higher confidence in the study area dataset, not necessarily in others. I'm sure there will metamorphic-dominated terranes where the subordinate igneous rocks will be classified with higher confidence.
→ We expounded on this in the updated version.

**Technical corrections**

line 44: delete ', particularly as it is likely to have been conducted by tens to hundreds of geologists...' with something like 'as all logging geologists have their own personal biases.
→ This was revised in the updated version.

Line 49-50: delete 'even detection of'
→ This was revised in the updated version.

line 57: The semi-automatic methods also are poor at describing textural characteristics (foliation, banding, grainsize variation)
→ This was revised in the updated version.

line 70" Delete 'Elizabeth'?
→ This was revised in the updated version.

line 105: limitations -> limitation
→ This was revised in the updated version.

line 198: replace 'that occurred' with 'were emplaced'
→ This was revised in the updated version.

line 199: replace 'ultramafic mafic' with 'ultramafic to mafic' and 'local centres' with 'local eruptive centres'.
→ This was revised in the updated version.

line 201: replace 'volcanoclastic' with volcaniclastic'
→ This was revised in the updated version.

line 203: delete 'profiles' and delete 'both'. Break sentence after 'bedrock' and start next with 'Regolith...'
→ This was revised in the updated version.

line 209: suggest replacing 'complexity' with 'diversity'
→ We changed it to "structural complexity". "Diversity" does not capture the complexity in relationship between these lithologies.

Figure 3 needs a unit to describe dill hole density e.g. per square kilometre
→ This was revised in the updated version.

line 254: Insert after 'Added records' 'with examples'
→ This was revised in the updated version.

line 270: Replace 'GeoSciML' with 'the CGI-IUGS Simple Lithology vocabulary
http://resource.geosciml.org/classifier/cgi/lithology'
→ This was revised in the updated version.

line 276: Suggest deleting second half of sentence i.e. after 'dictionary'
→ This was be revised in the updated version.

line 294: Delete orphan 'And'
→ This was revised in the updated version.

Figure 4: Lighten purple shade (or whiten text within)
→ This was revised in the updated version.

line 358: Replace 'Dip: it is the inclination angle perpendicular to the azimuth…' with 'Inclination: the plunge angle of the drill hole relative to horizontal…'.
→ This was revised in the updated version.

line 358-360: Replace sentence 'A positive value indicates an upward-directed drill hole and a negative value indicates a drill hole directed downwards.'
→ This was revised in the updated version.

line 411: Replace 'The string followed by key phrases such as…' with 'The string preceded by key phrases such as…'
→ This was revised in the updated version.

line 415: Does 'tokens with less than three characters' mean or include short words?
→ Yes, it does include 2 letter words. But most of two-letter words are prepositions (to, in, at, etc.). Only obvious issue should be "aa flows", which has not been observed as terminology used in the logs.

Line 434: Insert 'method' after 'ratio()'.
→ This was revised in the updated version.

Line 511: Font change in 'Company_Lithology'.
→ This was revised in the updated version.

Line 570: Where is the 'brown text' in Table 2?
→ This was revised in the updated version. We will symbolize it as bold to avoid confusion.

Figure 10: Lighten purple shade (or whiten text within).
→ This was revised in the updated version.

Line 598: replace 'take a look' with 'looked'
→ This was revised in the updated version.

Line 663: replace 'couple of' with 'four'
→ This was revised in the updated version.

Line 671: replace 'trumps' with 'trump'
→ This was revised in the updated version.

Line 833: What 'information being fed itself' mean?
→ This was revised in the updated version.

Line 887: delete 'a couple of'
→ This was revised in the updated version.

---

## Author Comment (AC2)

**Response to Ignacio Fuentes**

The breakdown of replies to the individual comments are below:

Abstract, Line 23: what is an extraction rate of 865? What units? Or is it a typo and should be 86.5%?
→ This should be 86%, this was revised in the updated version.

Introduction, Line 43: I'm not sure if lithological drill core logging is "inevitably" subjective. In my impression, a lack of standardization in the lithological descriptions makes it subjective, but the subjectivity might be reduced through a standard procedure of description. However, it is not clear that such standardization is what we want.
→ The information and level of detail contained in logs is highly dependent on the purpose of the study, this already makes geological logging subjective. This subjectivity is also influenced by the lack of standards between project and/or companies combined with the personal biases of the logging geologists.

Materials and methods, Line 146: "The module was re-written into python to be make it more compact"... The grammar there sounds funny.
→ This was revised in the updated version.

Materials and methods, Line 195: Figure 2 seems to be wrongly enumerated in the text. The study area is referring to Fig 2, but it corresponds to Fig 3.
→ This was revised in the updated version.

Materials and methods, Figure 4: Regarding volcaniclastic rocks, they are classified as igneous and sedimentary. Is it ok to have the same subgroup in two lithological groups?
→ The matching is done at the Detailed_Lithology level, thus not causing confusion in the Subgroup and Group level. Volcaniclastics are present in both lithological groups as although volcaniclastics are volcanic in origin and are categorized as igneous rocks, ambiguous lithologies such as "metavolcaniclastic_sandstone" is more sedimentary than igneous.

Materials and methods, Line 325: it should be EPSG:4326 for WGS84
→ This will be revised in the updated version.

Materials and methods, Line 326: Relative level with respect to the sea level? Does the relative level refers to any reference level or is it a standard level? Because if it refers to any reference level, there is no way to know the real location unless it is corrected using a DEM and assuming the collar at the surface of the terrain.
→ We use RL here to refer to elevations of survey points with reference to the mean sea level. This definition of RL is equivalent to the elevation values used in DEMs. It is possible to cross-check the values extracted to the values in the DEM. However, it is important to note that drillholes could be drilled from underground, thus would not have a collar location at the surface of the terrain. There are entries with positive survey dip/inclination values that suggest underground holes drilled upwards.

Results, Line 498: you specified 820,612 entries for lithology, and 273,684 matched records with the thesaurus. What happened with the remaining 546,819 entries (66.6% of the total entries)? Can you give a simple example of entries not matched?
→ We did not obtain a match with a score greater than 80. Example of unmatched entries in Table 2. Added this information the manuscript.

Results, Line 507: Does it mean that in about 40,000 records you had a ratio() score from the fuzzy string matching) lower than 80? Maybe you could be more explicit in this? Additionally, you defined

the score threshold based on the exact match. But, might it be a kind of balance between the number of matched records and the exact match percentage? I'm just wondering because it seems that by defining that threshold you lose a lot of entries to be matched (83.5%). Could you give a simple example of records not matched?

→ Yes, we expounded on this in the updated version. It is a balance, as you mention. In this case study, we selected a cut-off score of 80 since this is where the # of exact matches obtained plateaus. A lower cut-off score could be used, depending on the familiarity to the data and/or purpose of drillhole processing. For our case, we wanted to be as conservative as possible without being too stringent (cut-off score 100). Example of unmatched record in Table 2.

Results, Lines 542 - 560: These are more materials and methods than results.
→ This was revised in the updated version.

Results, Line 549: Couldn't you get the rest of the hierarchical categories based on the lower hierarchy defined?
→This is what we do, the Subgroup and Group level matched is based on the detailed lithology match.

Results, Table 2. It gives an example of unmatched cases, so disregard that part of previous comments.

Results, Lines 663 - 678. Accuracy metrics should be included in the materials and methods section and not in the results.
→ This was revised in the updated version.

Discussion, Lines 812: Did you tried to replace the "same as above" with the descriptions?
→ This was not included in the scope of this work. Replacing "same as above" requires building a dictionary for all possible permutations to refer to this (blank, ""). It also makes some assumptions that the rock type is exactly the same.

Discussion, Lines 824: Is there any way to automatise the building of a thesaurus given new advances in NLP and machine learning?
→This should be possible. However, there is a need to first understand the syntax in which geological data is captured. The thesaurus provided by dh2loop provides a starting point/training set for this.

Discussion, Line 831: "extraction rate of 16% from the Comments is not bad at all" This sounds too subjective, how do you define what is a good or a bad extraction rate?
→This was revised in the updated version by describing further that although it is a low extraction rate, there is value in being able to obtain 16% more data that was previously deemed "unusable". We also stated the number of records that amount to 16%.

Discussion, Lines 852-853: "For sedimentary rocks, the lack of a standard syntax as to how comments are recorded impacts the classification. " You see, imagine if such standardization is achieved, wouldn't it reduce the subjectivity
→ Standardization will definitely reduce subjectivity. However, achieving a standard is not an easy task, nor is it ours. This is something that is for the geological surveys to decide and implement. It is also important to note that a "standard" would be tricky to achieve as the information and level of detail contained in logs is highly dependent on the purpose of the study. This study provides a basis for creating a pre-standard. Not so much providing a guide of practice but highlighting what shouldn't be done and what practices create ambiguity.

Discussion, Line 863: grammar error " Soils are technically are not rocks"
→ This was revised in the updated version.

---

## Author Comment (AC3)

**General remarks**

The paper seems long. The length, and difficulty of feeling out the structure of a long paper, probably led to a feeling of confusion about whether I was reading a geological case study, a paper on a new approach to ingesting data, or the documentation for a software package. Perhaps the manuscript is trying to do too much? I wonder if you could split it into pieces? For example, move the more prosaic stuff (lists of tables, etc) to the docs; describe the more technical language modeling piece in a short nerdy paper about that; and present the case study as a short geological success story?

→ We agree that the paper is quite lengthy. However, splitting the content into multiple papers may not be as impactful. We addressed the sense of confusion by restructuring the paper into two main sections, the theory and the case study. The structure is as follows:

1. Introduction
2. dh2loop Drillhole Data Extraction
2.1    Conventions and Terminologies
2.2    Dependencies
2.3    Data Source
2.4    Thesauri
2.4.1   Drill Hole Lithology Codes Thesaurus
2.4.2   Clean-up Dictionary
2.4.3   Lithology Hierarchical Thesaurus
2.5    Data Extraction
2.5.1   Collar Extraction
2.5.2   Survey Extraction
2.5.3   Lithology Extraction
2.6    Fuzzy String Matching Assessment
3. Case Study: Yalgoo-Singleton Greenstone Belt
3.1    Study Area
3.2    Data Extraction Results
3.2.1   Collar
3.2.2   Survey
3.2.3   Lithology: Lithology Code Workflow and Comments Workflow
3.3    Fuzzy String Matching Results
3.3.1   Structure and Texture
3.3.2   Igneous Rocks
3.3.3   Sedimentary Rocks
3.3.4   Metamorphic Rocks
3.3.5   Surficial Rocks
4. Discussion
4.1    dh2loop Functions and Notebooks
4.2    Thesauri
4.3    Data Extraction
4.4    Assessment of String Matching Results

If the authors and editors feel a long paper is justified, then my suggestion is to be extra careful about the outline of the paper, so that a person knows what aspect each chunk of the paper is addressing.
→We agree. Please see previous response.

Compounding the length is that there's quite a bit of redundancy between text, figures and captions. Figure 9 is an archetypal example of this. I think the caption (which I cannot parse) essentially spells out the diagram (which I cannot read very well), and virtually all of the data is also written out in sections 3.1 to 3.3. My advice is to frame the ideas with the text and put all the details of table names and data in the figure, which should barely need a caption.
→We agree. We removed most of the redundant text and kept the information in the captions.

Some pieces seem better suited to the tool's documentation. Appendices B and C probably don't need to be part of a paper. Similarly, I think detailed descriptions of tables are clogging up the text and getting in the way of the reader. Standard measures like precision and recall, or string matching statistics, probably don't need to be described in detail.
→We addressed this and removed it from the Appendices. The standard measures were also removed.

**Detailed remarks**
- Title: Python should be written with an uppercase P.
  →This was revised in the updated version.
- Abstract: I'm not sure you need the first paragraph; it's introductory material. If you want to motivate the problem, I think you can do it in a sentence (the second sentence captures it for me).
  →This was revised in the updated version.
- Line 12: surveys, textual (no 'r')
  →This was revised in the updated version.
- Line 15: hundreds of
  →This was revised in the updated version.
- Line 19: replace 'that provides the functionality to extract...' with 'for extracting...'
  →This was revised in the updated version.
- Line 23: 86%
  →This was revised in the updated version.
- Line 51: You want spaces after all your semi-colons.
  →This was revised in the updated version.
- Section 2: I suggest sticking to sentence case for your subheadings; at least make them all the same.

→We made them all the same.
- Section 2.1, Conventions: The font is called Lucida, not Lucinda. I like the idea of a font convention, but if you're going to use one it is important to be consistent... and being consistent pretty hard. E.g. line 174: collar should be a table name, I think? Line 217: are these tables? Line 225: dh2loop should be italic.
  →We double checked the font convention to make sure none of it is missed.
- Lines 166–184: This paragraph is really hard to parse. I think the reader may start wondering if they need to know this information. If the info is important, I think you can let Figure 2 do the work.
  →This was removed from the text and kept in Fig 2.
- Line 280–295: Similarly, I think you're just describing Figure 4.
  →This was revised in the updated version.
- Sections 2.4.1, 2.4.2, 2.4.3: This feels like documentation.
  →These sections are kept but reduces in detail.
- Lines 419–440: There is a lot of information here; I feel it is perhaps best suited to documentation.
  →We decided to keep this section as it explains the string matching algorithm used by dh2loop which influences its results.
- Table 1: Not sure what to make of this. What do the checks and x's mean? What is the score? Why is one row bold?
  →This was revised in the updated version.
- Sections 3.1, 3.2, 3.3: See my earlier remarks about these paragraphs vs Figure 9 and its caption.
  →This was revised in the updated version.
- Table 2 and Figure 10: I find these hard to get insight from.
  →We decided to keep these.
- Confusion matrices: There are a great many data tables here but I am unsure what to do with this information. I see that there are a lot of under-represented labels (low support). The only reference I could find to this problem mentions normalizing the accuracy, but this doesn't really solve the problem, it just makes the confusion matrix colourbar fit better. So you're still unable to balance precision and recall (with small support, one of them is probably going to be bad). I'm not sure what you could do about it, other than get more data, but it might be worth mentioning.
  → We expounded on this. However, this is the nature of using real-word examples. Geologic logs have always been imbalanced as more detail is given to particular lithologies depending on the purpose of the study.

**Remarks about code**
There are some problems with the licences on some of the code you are using. **QDriller** (https://github.com/valheran/QDriller) is licenced under the GPL, which is a copyleft licence. This means that if you use it then your entire project must be licenced under the same (or a compatible) licence. **You are not allowed to use GPL'd code under an MIT licence.** So you have a few options:

1. If you keep that code as-is, you must licence your entire project under the GPL.
2. Ask the original author to grant you permission to re-use the classes you need under a more permissive licence. (Ideally, it would be, say, LGPL and a completely separate library that you could simply depend on, without copy/pasting the code across.)
3. Rewrite this functionality from scratch.

→This is not used in dh2loop and thus was removed from the repository.

**GeoVectoLitho** (https://github.com/IFuentesSR/GeoVectoLitho) is not licenced, as far as I can tell. Normally that would mean that you cannot use it without written permission granting you some rights. So make sure you have written permission to place it under your MIT licence, or you could ask the authors if they will consider an open licence. At the very least, you need to make it clear that the mlp.py module is based on work that is  copyright of those authors and explain the terms under which you're using it. You should probably also exclude it from the MIT licence so that nobody comes along and picks ut up thinking it's FOSS.

→The mlp.py code was used to compare the results with MLP. This has been removed from the repository.

In each of these cases, if you cannot find a way to

The Where to start instructions on the repo need VTK, folium and ipyleaflet adding to the installs. You could just tell people to install everything in requirements.txt with pip install -r requirements.txt. After that the notebooks ran for me (though I didn't run them all right through).

In Notebook 0, the map did not appear; I didn't try to figure it out. Enabling the extension didn't help. (I'm on a Mac, Chrome browser.)

→We revised this.

I'm not sure these comments should form part of a peer review. If you don't plan to maintain this library into the future, you can probably just ignore all this. But in case it's useful:

- I strongly recommend writing docstrings for all of your functions.
- You should write tests for your functions. "Untested code is broken code."
- You can remove components of the standard library — sys, math, re, time, itertools, etc — from requirements.txt. These are included in every Python installation.
- I advise against hard-coding the file encoding in your functions (with encoding = "ISO-8859-1") because this seems like something another person is quite likely to want to change, e.g. if they have a UTF-8 encoded file. You should expose it as an argument. The same goes for database connections and other things another user might want to change.
- You should delete unused code, rather than commenting it.

- I'm sure you know about geopandas already, but it seems like it would be useful in your project.
- Note that PyProj emits a warning in Python 3.8+ FutureWarning: '+init=:' syntax is deprecated.
- I don't know if psycopg2 connections or cursors are compatible with Python's context manager (in which case you should 'with' blocks for them), but even if they don't you should put them in try-finally blocks to ensure they get closed no matter what happens in the runtime.
- I recommend using a linter like flake8 to find things like multiple or redundant imports (e.g. you import numpy three times in dh2l_db.py). It will also help you adhere more closely to the PEP8 standard, which will make your code more readable and reusable.

Overall, the code has a non-Python feel to it (I don't know Fortran, but I've read a lot of code from geophysicists and they often write code like this!). Patterns like functions mutating global variables are not common in Python. Typically, one would pass the variable in to the function, then return them to the user (these are so-called 'pure functions'). You will find this easier to maintain. It feels strange to me to run a function like dh2l.litho_dico(litho_dic_file) and not have something (like a new DataFrame) returned to me.

→We will keep the code in current form and address these suggestions in a future release.

---

## Referee Report (RR1)

Thank you for the chance to see this revision.

I remain positive about the potential for this work, but I still regard this manuscript as too long. There is so much detail here that I find it difficult to appreciate the story. I'm sorry if this seems harsh, and possibly I'm in the minority in this feeling, but the purpose here is to get your research into the world so that people can read and appreciate it, not to document every aspect of the work.

So while I can see that the MS has been reduced by about 20%, I believe it could stand being half its current length. Most of what has been culled so far was the appendices, as far as I can tell. But I think there's scope for more brevity in the paper itself.

The good news is that I don't think it's difficult to make this paper a lot more readable. For example:

- The figure I picked on last time, Figure 9, is still repeated in its own caption, and with everything written out in Figure 3.2.1 and Figure 3.2.2.
- Have a look at the caption to Figure 3. I think it could be reduced to "Figure 3. Lithology Hierarchical Thesaurus showing the 7 major Lithology_Groups." I think there are a few pieces of information in the caption which are not simply repetition of what's in the diagram — but they are drowned out by everything else.
- As I also mentioned before, you do not need to provide the equations for standard statistics like precision and recall.

Turning to the code, I see you have addressed the licensing issues I mentioned (e.g. with QDriller). There are others however — I'm afraid my licence review was not that thorough. For example, you're depending on fuzzywuzzy — another GPL'd library. Again, this must be replaced with something compatible, or you must change your licence. I just had a quick look and couldn't see any others, but you should do a thorough audit here and be careful about using code from places like StackOverflow, most of which is also copyleft (CC BY-SA). (A side-note about your requirements.txt — you don't need to put things from the standard library in there, like math, sys, bisect, etc.)

I will mention that I do still have some reservations about the overall code quality (no docstrings, no tests, non-Pythonic idioms), but as I said before it's not clear to me how the journal feels about this sort of thing being part of a peer review.

I sincerely hope you persevere with this piece of work, because I do believe it deserves to be published.

---

## Author Response (AR2)

Dear Dr. Hall and Dr. Wickert,

Thank you for the comments and further direction on the revised paper. The main comment was on the length of the paper. We have managed to shorten it by a few pages by shortening the captions and moving the sections on collar and survey tables to the Appendix.

The issues on the software code will be handled by the group currently working on the further development of the code. Since it is not a requirement for GMD, we will keep it as is, for now.

Hope you find the submission in good order.
Thank you.

Warm regards,
Ranee Joshi